# A new state dependent parameterization for the free drift of sea ice

Charles Brunette[1], L. Bruno Tremblay[1], and Robert Newton[2]

[1]Department of Atmospheric & Oceanic Sciences, McGill University, Montreal, QC, Canada
[2]Lamont-Doherty Earth Observatory, Columbia University, Palisades, NY, USA

**Correspondence:** Charles Brunette (charles.brunette@mail.mcgill.ca)

**Abstract.** Free drift estimates of sea ice motion are necessary to produce a seamless observational record combining buoy and satellite-derived sea ice motion vectors. We develop a new parameterization for the free drift of sea ice based on wind forcing, wind turning angle, sea ice state variables (thickness and concentration) and estimates of the ocean currents. Given the fact that the spatial distribution of the wind-ice-ocean transfer coefficient has a similar structure to that of the spatial distribution of sea ice thickness, we take the standard free drift equation and introduce a wind-ice-ocean transfer coefficient that scales linearly with ice thickness. Results show a mean bias error of -0.5 cm/s (low-speed bias) and a root-mean-square error of 5.1 cm/s, considering daily buoy drift data as truth. This represents a 35% reduction of the error on drift speed compared to the free drift estimates used in the Polar Pathfinder dataset (Tschudi et al., 2019b). The thickness-dependent transfer coefficient provides an improved seasonality and long-term trend of the sea ice drift speed, with a minimum (maximum) drift speed in May (October), compared to July (January) for the constant transfer coefficient parameterizations which simply follow the peak in mean surface wind stresses. Over the 1979-2019 period, the trend in sea ice drift in this new model is +0.45 cm/s decade$^{-1}$ compared with +0.39 cm/s decade$^{-1}$ from the buoy observations, whereas there is essentially no trend in a free drift parameterization with a constant transfer coefficient (-0.09 cm/s decade$^{-1}$) or the Polar Pathfinder free drift input data (-0.01 cm/s decade$^{-1}$). The optimal wind turning angle obtained from a least-squares fitting is 25°, resulting in a mean error and a root-mean-square error of +3° and 42° on the direction of the drift, respectively. The ocean current estimates obtained from the minimization procedure resolve key large scale features such as the Beaufort Gyre and Transpolar Drift Stream, and are in good agreement with ocean state estimates from the ECCO, GLORYS and PIOMAS ice-ocean reanalyses, and geostrophic currents from dynamical ocean topography, with a root-mean-square difference of 2.4, 2.9, 2.6 and 3.8 cm/s, respectively. Finally, a repeat of the analysis on two sub-sections of the time series (pre- and post-2000) clearly shows the acceleration of the Beaufort Gyre (particularly along the Alaskan coastline) and an expansion of the gyre in the post-2000s, concurrent with a thinning of the sea ice cover and the observed acceleration of the ice drift speed and ocean currents. This new dataset is publicly available for complementing merged observations-based sea ice drift datasets that include satellite and buoy drift records.

## 1 Introduction

Communities living in Arctic regions have had an implicit understanding of the drift of sea ice for many hundreds of years, with the sea ice playing a major role in the way of life (Aporta, 2002; Krupnik et al., 2010). Describing sea ice dynamics around Igloolik, Nunavut, Inuit elder Aipilik Inuksuk recalls: "Sometimes when there are strong winds, the new ice and the land-fast

ice cannot come in contact with each other because the northerly winds cause the newly formed ice to break up and drift away. After the winds die down and the weather improves, the resultant open water freezes again and the current will move the new ice back and forth against the land-fast ice. [...] This is true, and that is the nature of the moving ice" (Inuksuk, 2011). That is

to say that for a very long time, people have been aware that sea ice circulation is mainly driven by surface stresses from the atmosphere and the ocean. Over the last century, from the 1893-96 Nansen drift aboard the Fram to the 2006-07 TARA and and 2019-20 MOSAiC expeditions, polar oceanographers have tried to break down the processes relating sea ice motion to winds and oceanic currents by means of observational campaigns and theoretical development. The relationship of ice motion to external stresses is complicated by the internal rheology of the ice pack: the way in which ice resists or deforms, rather than

moving when external stress is applied. In decomposing ice drift, one useful simplification is to assume that the ice is free to drift in response to the wind, i.e. there is no significant impact from the internal rheology. Such a free drift approximation is used in sea ice tracking models wherever the only data input is the wind field (Tschudi et al., 2019b; Krumpen, 2018; Campbell et al., 2020) In this contribution, we propose a new parameterization for estimating sea ice motion based on free drift, including a dependency on thickness as a state variable.

Tracking sea ice motion in the Arctic can support a wide range of studies, including to: calculate sea ice age (Tschudi et al., 2019a); quantify changes in the dynamic response of a thinner and less compact ice pack under climate change (Mahoney et al., 2019; Belter et al., 2020); investigate mechanisms for seasonal forecasting of sea ice based on late-winter off-shore ice motions, at the pan-Arctic scale or at regional scales (Nikolaeva and Sesterikov, 1970; Krumpen et al., 2013; Williams et al., 2016; Brunette et al., 2019; Kim et al., 2021); inform socio-environmental studies by quantifying pollutant or phytoplankton

transport by sea ice between different peripheral seas (Newton et al., 2017; Lind et al., 2018) and paleoclimate studies, by identifying provenance of ice-rafted sediment to infer past sea ice drift motion and conditions (Darby, 2008; Polyak et al., 2010; Tremblay et al., 2015). Spatially and temporally complete ice motion datasets crossing the summer satellite data-desert are essential for these applications. One key contribution of this study is to produce a free drift product with documented errors (both spatially and temporally) that spans the season when satellite-based drift estimates are sparse. The hope is that

this will encourage a wider range of independent, seamless ice motion datasets, also covering the summer period which is oftentimes avoided because of larger error in drift estimates from passive microwave. Such a dataset is also included in the Sea Ice Tracking Utility, made publicly available recently on the National Snow and Ice Data Center website (SITU, Campbell et al. 2020), or the Alfred Wegener Institute ICETrack tool (Krumpen, 2018) – for educational, scientific and field expedition planning purposes.

Free drift estimates of sea ice motion are useful to complement other observational ice motion products. GPS-equipped drifting buoys remain the most accurate source of ice drift information, but are limited in space and time. For instance, the data record from the International Arctic Buoy Program (IABP) is composed of over 1920 buoys since 1979; but there are only a few tens to a little over a hundred buoys present at any one time. Remote sensing of the cryosphere, mainly from satellite passive microwave or synthetic aperture radar instruments, provides Arctic-wide observations of the sea ice surface, from which ice

motion can be derived using different image-processing algorithms (Emery et al., 1995; Kwok et al., 1998; Meier and Dai, 2006; Lavergne et al., 2010; Tschudi et al., 2010). Reliable satellite-derived drift vectors are more abundant in the winter, but

much more sparse in the summer, when clouds and melt-ponds affect passive microwave retrieval (Sumata et al., 2014). When neither buoy nor satellite information is available, estimates of the ice motion in response to the wind fields are essential to "fill the gap" and maintain a spatially and temporally complete ice motion dataset, such as the National Snow and Ice Data Center's Polar Pathfinder (Tschudi et al., 2019b).

Free drift is defined as the motion of sea ice in response to atmospheric and oceanic forcing in the absence of internal ice stresses. We can write the steady-state free drift $U_i$ as a linear function of the wind velocity $U_a$, i.e. $U_i = \alpha U_a + U_w$, where $\alpha$ is an integrated wind-ice-ocean transfer coefficient, and the term $U_w$ is the ocean current. $\alpha$ is a complex coefficient that represents the magnitude of the momentum transfer from the wind to sea ice relative to the ocean current. This wind-ice-ocean transfer coefficient includes a scaling factor $|\alpha|$, also referred to as 'wind factor' in the literature, expressed as a percentage from wind speed [m/s] to ice drift speed [cm/s], and a turning angle $\theta$ that estimates the turning angle between wind forcing and the sea ice response, due to the rotation of the Earth: $\alpha = |\alpha|e^{-i\theta}$. An estimate of this coefficient derived from the historical Fram expedition gives a value of $|\alpha| \approx 2\%$ with a turning angle of $\theta = 20 - 40°$ to the right of the near-surface winds (Nansen, 1902). When considering the surface geostrophic wind instead, Thorndike and Colony (1982) report values of $|\alpha| = 0.8\%$ and $\theta = 5°$ for the fall, winter and spring, and $|\alpha| = 1.1\%$ and $\theta = 18°$ for the summer. Note that the winds veer to the left in the atmospheric boundary layer due to surface friction and the ice drifts to the right of the surface winds, leading to smaller turning angles when the geostrophic wind is considered (Leppäranta, 2011). Using geostrophic winds and buoy drift data over the 1979-1993 period, Thomas (1999) retrieved typical values for the wind-ice-ocean transfer coefficients (turning angle) of 0.7% (0°) for the winter season (defined as November-April), and 1.1% (18°) for the summer (defined as June-September). Analyzing six sea ice buoys and 10 m wind data from a meteorological station in the Baltic Sea, Uotila (2001) reports a transfer coefficient (and turning angle) in the 1.3-3.3 % (23-25°) range. Citing Thorndike and Colony (1982), the Polar Pathfinder sea ice motion dataset uses a constant coefficient (and turning angle) of 1% (20°) with surface geostrophic winds derived from the National Centers for Environmental Prediction / National Center for Atmospheric Research (NCEP/NCAR) reanalysis (Kalnay et al., 1996).

Arctic wind speeds are larger in winter compared to summer (with a maximum spanning December-January-February and a minimum in May-June-July, based on the European Center for Medium-Range Weather Forecasts Atmospheric Reanalysis 5th Generation, ERA5), while the maximum sea ice drift occurs in September-October and the minimum in March. Thicker and more compact sea ice, typical of late winter conditions, results in stronger ice interactions within the ice pack, reducing the ice drift for a given wind speed, and resulting in a minimum in sea ice drift speed in the late winter, despite the winds being at their seasonal maximum (Olason and Notz, 2014; Tandon et al., 2018; Yu et al., 2020). The offset between the seasonal cycles of wind speed and ice drift speed indicates that sea ice state parameters are essential for describing the seasonally varying drift of ice motion. This can also be understood in terms of an energy balance for sea ice, where the power input from the surface wind stress is mainly dissipated by the water drag as well as the internal ice stresses in compact ice regions, when ice interactions are important (Bouchat and Tremblay, 2014). Mapping the distribution of the wind-ice-ocean transfer coefficient (based on passive microwave-derived ice drift and geostrophic winds over the mid-1990s), Kimura and Wakatsuchi (2000) report a sharp contrast between seasonal ice zones, such as the Bering, Barents and Okhotsk Seas, where the value can reach

2%, and the Arctic interior where the value drops to 0.8% and below; a spatial pattern which, they hypothesize, relates to stresses internal to the ice pack, and therefore to ice thickness and concentration. The same spatial pattern is observed over the 2003-2017 period by Maeda et al. (2020), who additionally report on the seasonal cycle of the transfer coefficient (minimum around 0.8% in March and maximum around 1.1% in October), and on positive long-term trends across the whole Arctic. Interestingly, Maeda et al. (2020) note that the upward trend of the transfer coefficient stops after 2010, particularly in regions where multi-year ice used to be prevalent. The marked differences in the seasonality of the surface wind stress and ice drift calls for a parameterization of the wind-ice-ocean transfer coefficient that also considers ice state variables. The goal of this work is to develop such a parameterization for the transfer coefficient ($\alpha$). To the best of our knowledge, this has not been attempted before.

The proposed parameterization for the wind-ice-ocean transfer coefficient $\alpha$ is akin to the efforts of Tremblay and Mysak (1997); Steiner (2001); Lu et al. (2011); Tsamados et al. (2014); Lüpkes and Gryanik (2015); Toyoda et al. (2021), among others, who developed ice state dependent parameterizations for the ice-atmosphere and ice-ocean drag coefficients that are included in the momentum balance equations for numerical sea ice models. Tremblay and Mysak (1997) show that model runs including drag coefficients that depend linearly on ice thickness are in better agreement with observations. In their ice-ocean model, Toyoda et al. (2021) achieve a 15-20% error reduction on the sea ice velocity fields by introducing parameterizations for drag coefficients, the ice-ocean turning angle, and the ice pressure parameter. Castellani et al. (2018) demonstrate that introducing variable drag coefficients in the ice-ocean coupled configuration of the MITgcm model improves the realism of the simulation. A theoretical study by Lu et al. (2016) explores the sensitivity of the scaling factor $\alpha$ and the turning angle $\theta$ to variable drag coefficients dependent on ice concentration and ice floe geometry, in a free drift ice motion regime. The current study builds on the work of Lu et al. (2016) – who only considered a range of sea-ice concentration between 0 and 80 %, limiting the applications of their parameterization to the marginal ice zone. In contrast with these previous studies, we propose a simpler approach, in which the transfer coefficient $\alpha$ is a function of the ice state, in the formulation for the free drift of sea ice. A state-dependent $\alpha$ can be conceptually understood as an integrated metric taking into account the spatial and temporal variability of both the atmosphere and ocean drag coefficients.

Another challenge in estimating free drift ice motion vectors arises from the poorly constrained Arctic ocean surface currents under sea ice. Existing observational approaches to estimate surface currents in the Arctic Ocean include i) the use of ocean dynamic height derived from satellite altimetry, from which geostrophic currents can be derived (Armitage et al., 2017), ii) direct current measurement from ice-tethered acoustic Doppler current profiler (ADCP, McPhee, 2013), iii) using wind stress data and depth-integrated vorticity balance (Nøst and Isachsen, 2003) and iv) deriving the mean surface ocean circulation from time-averaged sea ice drift data (Thorndike and Colony, 1982; Kimura and Wakatsuchi, 2000; Kwok et al., 2013). We expand on the approach of Thorndike and Colony (1982) for estimating the surface oceanic currents, using drifting buoys and wind data to produce updated estimates of the surface oceanic circulation in the Arctic.

This paper is structured as follows. Section 2 introduces the buoy, sea ice, atmospheric and oceanic datasets. Section 3 describes the methodology for parameterizing of the wind-ice-ocean transfer coefficient $\alpha$. Section 4 presents the new estimates

of free drift sea ice motion and quantifies the error with respect to buoy data. Section 5 summarizes the main findings presented in the paper.

## 2 Data

### 2.1 Grid

We use a 25 km Equal Area Scalable Earth grid (EASE grid, Brodzik and Knowles, 2002) as a common grid for all datasets. The advantage of using the EASE-grid is that all grid cells have the same area and roughly the same length in x and y. This facilitates working with vector quantities such as sea ice, ocean, or wind velocities. The u- and v-components of velocity are relative to the x- and y-directions in the EASE-grid frame of reference – i.e. they do not represent zonal and meridional velocities. All fields not natively gridded on the EASE have been interpolated to the 25 km EASE-grid using a Delaunay triangulation interpolation

approach, which builds on piece-wise linear interpolation of scattered data. We work with the original EASE-grid – as opposed to the more recent EASE-grid 2.0 – to maintain continuity with current products distributed by the National Snow and Ice Data Centre (NSIDC) and other sea ice tracking utilities that were built on the original definition of the EASE-grid.

### 2.2 Sea ice buoys

We use the daily buoy drift vector product included in the Polar Pathfinder dataset, distributed by the NSIDC Tschudi et al.,

2019b. The drift vectors are calculated from the average of the offset 24-hour ice motion (midnight to midnight and noon to noon) from buoys in the IABP dataset (International Arctic Buoy Programme, 2020). In the following, we consider only buoy velocity estimates that are located within the ice edge, defined as where sea ice concentration is higher than 15%. A total of 457,915 drift vectors are available over the 1979-2019 period, from 1920 different Arctic drifting buoys and other instruments. The drift vectors distributed by the NSIDC are stored on a 25 km EASE-grid. The estimated error on the buoy position and

velocity are respectively under 300 m and 1 cm/s (Walt Myers, personal communication). In the following, we consider the ice motion vectors derived from buoy data as 'truth' when assessing the errors in the proposed free drift paramaterizations.

### 2.3 Polar Pathfinder free drift input data

The Polar Pathfinder dataset consists of merged sea ice motion vectors derived from buoy, satellite, and free drift estimates. The NSIDC also provides the raw motion vectors from each data product. In the following, we use the free drift raw vectors

processed by the NSIDC – equal to 1% of the geostrophic wind speed from the NCEP/NCAR atmospheric reanalysis (2.5° resolution), with a 20° wind turning angle. As a benchmark, we use the daily averaged values of the Polar Pathfinder free drift motion vectors, stored on a 50 km EASE-grid in a checker board pattern, then interpolated to a 25 km resolution. The reported root-mean-square error on the free drift motion estimates is 6.1 cm/s from a comparison with buoys (Tschudi et al., 2019b).

## 2.4 Surface winds

We use the 10m winds from ECMWF Atmospheric Reanalysis 5th Generation (ERA5) stored on a $1/4°$ grid (Hersbach et al., 2020) over the 1979-2019 period. Daily averaged wind velocities are calculated from hourly wind velocities. ERA5 is the reanalysis that performed best with respect to the 10m winds in a comparison of six different atmospheric reanalyses with observations from Norwegian Young Sea Ice 2015 campaign (N-ICE2015), Graham et al. (2019), with Winter/Spring/Summer correlations coefficients of 0.92/0.91/0.97, biases of +0.4/+0.1/-0.2 m/s and root-mean-square errors of 1.4/1.1/0.9 m/s.

## 2.5 Sea ice concentration

We use the National Atmospheric and Oceanic Administration / National Snow and Ice Data Center (NOAA/NSIDC) Climate Data Record (CDR) of passive microwave sea ice concentration version 3 (Meier et al., 2017), over the 1979-2019 period. The CDR sea ice concentration data is stored on a stereographic cartesian grid at a 25 x 25 km resolution. Specifically, we use the Goddard merged variable, a daily product based on the highest concentration value from either the NASA Team algorithm (Cavalieri et al., 1984) or the Bootstrap algorithm (Comiso, 1986). Earlier reports by Andersen et al. (2007) on different types of passive-microwave concentration retrieval indicate errors of 5% in the winter. More recently, NSIDC-CDR specific investigations by Kern et al. (2019, 2020) find a slight bias-high of +1% to +3.5% with a standard deviation of 5% for winter-type (100% ice concentration) ice conditions; while in the summer, the error is found to be much larger, with a standard deviation reaching up to 35%, and a mean overestimation on the order of 5 to 10%, when compared to MODIS-derived concentration. We use the sea ice concentration data to discriminate between motion vectors from buoys located on ice or in open water.

## 2.6 Sea ice thickness

Daily sea ice thickness is taken from the Pan-Arctic Ice Ocean Modeling and Assimilation System (PIOMAS) ice volume reanalysis (Zhang and Rothrock, 2003). PIOMAS is a coupled ice-ocean model, forced with atmospheric fields from the NCEP/NCAR reanalysis, that assimilates sea ice concentration from the near real-time NSIDC product based on the NASA Team algorithm (Lindsay and Zhang, 2006). PIOMAS captures the large scale structures of ice thickness in the Arctic; while being biased thick in thin ice regions and biased thin in thick ice regions (Schweiger et al., 2011). PIOMAS data is available in near-real time year-round for the full Arctic, and is stored on a generalized curvilinear coordinate system grid configuration with the pole shifted over Greenland.

## 2.7 Ocean currents

We consider four different ocean velocity products for comparison with the surface current estimates developed in this study. The Estimating the Circulation and Climate of the Ocean dataset (ECCO, Fukumori et al. 2019) covers the 1992-2017 period and is based on MITgcm model runs that assimilate a suite of ocean and sea ice observations at a 1/4° resolution. We use monthly averaged values of the u-component and v-component of the ocean velocity for the topmost level (5m depth). The

GLORYS ocean state estimates (Mercator Ocean, 2017) runs at 1/4°, is based on a global configuration of the NEMO OGCM coupled to the Los Alamos sea ice model CICE, and assimilates observations including temperature and salinity profiles, satellite sea surface temperatures, and satellite altimetry. GLORYS is available at a daily resolution from 1993 onward. We also retrieve the oceanic currents from the PIOMAS reanalysis (Zhang and Rothrock, 2003). The PIOMAS ocean currents are produced alongside the sea ice volume reanalysis. PIOMAS daily ocean velocity vectors (at a depth of 7.5 m) are available over the 1979-2015 period. Lastly, we use the geostrophic ocean currents from the Centre for Polar Observation and Modelling (CPOM), which are derived from dynamic ocean topography (DOT) based on satellite altimetry (Armitage et al., 2017) – referred to as the CPOM/DOT dataset from hereon. The monthly averaged geostrophic currents are available at a 0.75° x 0.25° resolution from 2003 to 2014, covering 60°N to 81.5°N. For each of these ocean velocity products, we calculate a climatology over the common 2003-2014 period.

## 3    Free drift of sea ice

### 3.1    Momentum equation for sea ice

The conservation of momentum for sea ice can be written as (Hibler, 1979):

$$\rho_i h_i \frac{d\boldsymbol{U}_i}{dt} = \boldsymbol{\tau}_a - \boldsymbol{\tau}_w - \rho_i h_i f \hat{k} \times \boldsymbol{U}_i + \nabla \cdot \boldsymbol{\sigma} - \rho_i h_i g \nabla H_d, \tag{1}$$

where $\rho_i$ is the ice density, $h_i$ the ice thickness, $\boldsymbol{U}_i$ is the ice motion vector, $\boldsymbol{\tau}_a$ is the surface wind stress, $\boldsymbol{\tau}_w$ is the ocean drag, $f$ is the Coriolis parameter, $\nabla \cdot \boldsymbol{\sigma}$ is the internal ice stresses term, $g$ is the gravitational constant, and $H_d$ is the sea surface dynamic height. The atmospheric stress and ocean drag are represented by quadratic drag laws:

$$\boldsymbol{\tau}_a = \rho_a C_a e^{-i\theta_a} |\boldsymbol{U}_a| \boldsymbol{U}_a, \tag{2}$$

$$\boldsymbol{\tau}_w = \rho_w C_w e^{-i\theta_w} |\boldsymbol{U_i} - \boldsymbol{U_w}| (\boldsymbol{U_i} - \boldsymbol{U_w}), \tag{3}$$

where $\rho_a$ and $\rho_w$ are the air and water densities, $C_a$ and $C_w$ are the air-ice and ice-water drag coefficients, $\theta_a$ and $\theta_w$ are the air and water turning angles, $\boldsymbol{U_a}$ is the 10m wind velocity and $\boldsymbol{U_w}$ is the ocean current. The ice velocity is neglected in the surface air-ice stress (Eq. 2) because $\boldsymbol{U_i}$ is typically much smaller than $\boldsymbol{U_a}$. Assuming steady-state, thin ice and neglecting internal ice stress, the main balance of terms in the momentum equation become: $\boldsymbol{\tau_a} = \boldsymbol{\tau_w}$, or:

$$\rho_a C_a e^{-i\theta_a} |\boldsymbol{U}_a| \boldsymbol{U}_a = \rho_w C_w e^{-i\theta_w} |\boldsymbol{U}_i^{fd} - \boldsymbol{U}_w| \ (\boldsymbol{U}_i^{fd} - \boldsymbol{U}_w), \tag{4}$$

where $\boldsymbol{U}_i^{fd}$ is the free drift ice velocity. In this simple case, an analytical solution for the ice velocity can be written as a function of the wind and ocean current velocities and a transfer coefficient $\alpha$ (Leppäranta (2011)):

$$\boldsymbol{U}_i^{fd} = \alpha e^{-i\theta} \boldsymbol{U}_a + \boldsymbol{U}_w, \tag{5}$$

where $\theta$ is an integrated turning angle that takes into account the turning in the surface ocean Ekman layer and (indirectly) the Coriolis effect, and where

$$\alpha = \sqrt{\frac{\rho_a C_a}{\rho_w C_w}}. \tag{6}$$

$\alpha$ includes both the air-ice and ice-water drag coefficients; it is technically a wind-ice-ocean transfer coefficient, even though it is applied only on the wind velocity. For simplicity, we refer to $\alpha$ as the transfer coefficient. This simple relation explains roughly 70% of the variability in sea ice velocity in the central Arctic (Thorndike and Colony, 1982). Limiting cases include low wind speed, strong Coriolis effect due to thick ice, and non-negligible internal ice stresses (Thorndike and Colony, 1982; Bouchat and Tremblay, 2014).

## 3.2 Free drift parameterizations

Our goal is to derive a free drift parameterization that is bias-corrected with respect to the buoy drift data from IABP - and that is sea ice state-dependent in order to take into account the seasonality and long-term trends in the sea ice drift, which are directly related to seasonal changes in sea ice thickness. The long-term changes in sea ice thickness associated with global warming are of the same order of magnitude as the seasonal change (Rothrock et al., 2008; Rampal et al., 2009), which implies that there

will be a trend in $\alpha$ from the parameterization as well. For clarity, we use subscripts for denoting the transfer coefficients $\alpha$ and wind turning angle $\theta$ in the different parameterizations of free drift:

- $\alpha_p, \theta_p$: Polar Pathfinder free drift input (Tschudi et al., 2019b);

- $\alpha_0, \theta_0$: constant transfer coefficient, no ocean currents;

- $\alpha_w, \theta_w$: constant transfer coefficient, including ocean currents;

- $\alpha_{ij}, \theta_{ij}$: spatially-varying transfer coefficient and turning angle (static in time);

- $\alpha_h, \theta_h$: thickness-dependent parameterization;

as well as an additional free parameter $\beta_h$ for the thickness parameterization (see section 4.3).

## 3.3 Minimization procedure

For the $\alpha_0$, $\alpha_w$, $\alpha_{ij}$ and $\alpha_h$ parameterizations, we use a least squares minimization approach to find the coefficients $(\alpha, \theta)$ and

ocean currents ($\boldsymbol{U}_w$) that will minimize the error function:

$$E_{\alpha,\theta} = \sum_{k=1}^{n} [\boldsymbol{U}_{buoy}^k - \boldsymbol{U}_{fd}^k(\alpha, \theta)]^2 \tag{7a}$$

$$E_{\boldsymbol{U}_w} = \sum_{k=1}^{n} [\boldsymbol{U}_{buoy}^k - \boldsymbol{U}_{fd}^k(\boldsymbol{U}_w(x,y))]^2 \tag{7b}$$

where $n$ is the total number of observational points, $\boldsymbol{U}_{buoy}$ are the daily buoy drift observations, and the free drift velocities $\boldsymbol{U}_{fd}$ are estimated at the same location and time as the buoy observations. Note that $\alpha$ can be either a constant or parameterized as a function of sea ice state variables. We solve the minimization problem using the Levenberg-Marquardt least-squares algorithm (MATLAB *lsqcurvefit* function). The error function is first differentiated with respect to each of the free parameters, and the resulting system of linear equations is solved iteratively using a combination of Gauss-Newton and steepest descent methods. In a first step (Eq. 7a), the $\boldsymbol{U}_w$'s are considered known, and we find the free parameter(s) $\alpha$ and $\theta$ that minimize the error function over all available data ($n = 457,915$). In a second step (Eq. 7b), the transfer coefficient(s) are fixed and we solve for the time-constant but spatially varying ocean currents $\boldsymbol{U}_w(x, y)$. The number of data points that go into the evaluation of each value of $\boldsymbol{U}_w(x, y)$ varies spatially (see Fig. 1). This iterative procedure is repeated until convergence. Assuming that the component of ice motion that is not explained by the winds (and the influence of ice thickness) can be attributed to ice-ocean drag, we assign the residual of the linear regression of the free drift transfer function – or in other terms the 2-dimensional intercept in the best linear fit – to the ocean current vectors, expanding on the work of Thorndike and Colony (1982).

The full convergence of the solution is reached within 5 iterations. Using this iterative procedure reduces the size of the matrix passed to the least-squares solver, and therefore greatly reduces the memory requirement. The initial guess for the iterative procedure is a value of one (unit-value) for all free parameters. The final solution (i.e. the minimum in the error function) is independent of the initial guess and the resulting transfer coefficients ranges from 1-2.5% (in line with previous estimates) depending on whether $\alpha$ is constant or sea ice state dependent.

The same procedure is applied to all of the free drift parameterizations, with the exception of the $\alpha_{ij}$ parameterization where the minimization problem is solved locally, for all of the $\alpha$, $\theta$ and $\boldsymbol{U}_w$ free parameters (see section 4.3).

The density of the data is such that there is insufficient information for interannually-varying ocean current fields. Therefore, we retrieve a 40-year mean climatology of the ocean current field. In the evaluation of the ocean currents, we use a 3x3 grid-cell (75 km x 75 km) search window centered on the target grid cell $i, j$ to have enough data points for the minimization procedure (Fig. 1). The size of this search window was defined such that at least 10 data points are present for each of the grid cell in the domain. This allows for estimates of ocean currents covering most of the pan-Arctic domain (with the exception of parts of the Eurasian plateau where buoy data is not included in the IABP dataset). The averaging introduces a smoothing of surface current features over a scale of 75 km and allows for the representation of large-scale features such as the Beaufort Gyre and Transpolar Drift Stream. This is not considered an issue in the context of long time-averaged fields.

## 4 Results and discussion

### 4.1 Bias-corrected free drift

In the following, error metrics are calculated by comparing the different free drift parameterizations to the IABP buoy daily drifts (Table 1). We use the NSIDC Polar Pathfinder free drift input data (Tschudi et al., 2019b) as a benchmark to evaluate the improvement attributable to the other parameterizations. For the Polar Pathfinder free drift, the root-mean-square error (mean bias error) on the norm of the velocity, u- and v- components are 7.9 (-4.0), 5.7 (0.2) and 5.7 (-1.2) cm/s, respectively. The bias-

**Table 1.** Values obtained from a least squares fit using buoy data (IABP), 10m wind velocity (ERA5) and ice thickness (PIOMAS) over the 1979-2019 period (with the exception of $\alpha_p$ for which the values are prescribed). These parameterizations include the Polar Pathfinder free drift input data ($\alpha_p$); a bias corrected, constant wind-ice-ocean transfer coefficient, ignoring ocean currents ($\alpha_0$) and permitting ocean currents ($\alpha_w$); spatially varying transfer coefficients, turning angles and ocean currents ($\alpha_{ij}$); and a thickness-dependent free drift parameterization ($\alpha_h$). The five rightmost columns present the root-mean-square error (RMSE, bold font), and mean bias error (in parenthesis) on drift speed, u- and v-components, drift angle, and the explained variance ($R^2$), evaluated at the IABP buoy data locations.

| | | Coefficients | | | Error metrics | | | | |
| | | | | | **RMSE** and (Mean bias error) | | | | |
| Name | Parameterization | $\alpha$ [%] | $\beta$ [1/m] | $\theta$ [°] | $\|U_i\|$ [cm/s] | $u_i$ [cm/s] | $v_i$ [cm/s] | $\theta$ [°] | $R^2$ |
|---|---|---|---|---|---|---|---|---|---|
| $\alpha_p$ | $\alpha_p e^{-i\theta_p}$ | 1.0 | - | 20 | **7.9** (-4.0) | **5.7** (-0.2) | **7.5** (-1.2) | **56** (-17) | 0.15 |
| $\alpha_0$ | $\alpha_0 e^{-i\theta_0}$ | 1.4 | - | 23 | **6.2** (-0.8) | **5.3** (0.0) | **5.1** (-1.1) | **45** (2) | 0.33 |
| $\alpha_w$ | $\alpha_w e^{-i\theta_w}$ | 1.3 | - | 24 | **5.4** (-0.4) | **4.8** (0.0) | **4.4** (0.0) | **42** (3) | 0.48 |
| $\alpha_{ij}$ | $\alpha_{ij} e^{-i\theta_{ij}}$ | Fig.4a | - | - | **4.8** (-0.5) | **4.4** (0.0) | **4.1** (0.0) | **42** (3) | 0.59 |
| $\alpha(h)$ | $\alpha_h(1-\beta_h h_i)e^{-i\theta_h}$ | 2.0 | 0.17 | 25 | **5.1** (-0.5) | **4.6** (0.0) | **4.3** (0.0) | **42** (3) | 0.55 |

corrected free drift parameterization ($\alpha_0$) with a constant transfer coefficient and no ocean currents – one that minimizes the error function above (Eq. 7a), as opposed to a 1% value for $\alpha$ and 20° value for $\theta$ – reduces the root-mean-square error (mean bias error) to 6.2 (-0.8), 5.3 (0.0) and 5.1 (-1.1) cm/s for the norm of the velocity and the u- and v- components, respectively (see Fig. 2). Note that the minimization procedure minimizes the cost function based on the u- and v-components of the drift simultaneously. One consequence of this is that there can be a residual mean bias error on the speed (a nonlinear function of u and v). When considering non-zero ocean currents ($\alpha_w$), the root-mean-square error and mean bias error are further reduced to 5.4 (-0.4), 4.8 (0.0) and 4.4 (0.0) cm/s for the norm of the velocity, the u- and v- components. Regarding the error on the drift direction (using circular statistics), the bias-corrected $\alpha_w$ parameterization with ocean currents reduces the root-mean-square and mean bias error to 45° (2°) compared to 56° (-17°) for the Polar Pathfinder free drift input, where a positive value in mean bias error indicates that the drift estimate is to the right of the buoy drift (Fig. 3c,f). To summarize this first step: there is a 22% reduction of the drift root-mean-square error and 80% reduction of the mean bias (from $\alpha_p$ to $\alpha_0$) by minimizing a cost function for the transfer coefficient and angle, and a additional 10% reduction of the root-mean-square error and 5% reduction of the mean bias by taking into account the ocean current as the intercept in the linear fit between the surface wind stress and ice drift speed. This result is in agreement with a single-drifter analysis of Uotila (2001), where the addition of ocean currents along a buoy track reduces by 18% the mean velocity difference between simulated and observed velocities. We note that the use of 10 m winds from the 1/4° ERA5 reanalysis reduces the root-mean-square error on the drift speed from 5.7 cm/s to 5.4 cm/s when compared with the 1° ERA-Interim reanalysis (results not shown).

The error reduction can be clearly seen in the distribution of ice drift speed (Fig. 3b,e), where the constant $\alpha_w$ parameterization yields a distribution that is in better agreement with that of the buoys, compared to the standard parameterization $\alpha_p$.

A peak at zero velocity is present in the IABP buoy drift speed distribution, which, aside from low winds, can be attributed to the presence of landfast ice, or thick ice north of the Canadian Arctic Archipelago, or more generally, strong internal ice interactions. The constant parameterization including ocean currents $\alpha_w$ also contributes to reducing the relative error (defined as the root-mean-square error divided by the mean). The bias-corrected $\alpha_w$ parameterization reduces the seasonal relative error by 23% and the interannual relative error by 22% compared to the standard $\alpha_p$ parameterization (Fig. 5c,d). However, the $\alpha_w$ parameterization does not improve by much the explained variance of the interannual variability of the drift speed (Fig. 5b): the $R^2$ (adjusted $R^2$) is 0.28 (0.26) for $\alpha_p$, and 0.23 (0.21) for the $\alpha_w$ parameterization, which is expected since $\alpha_w$ does not take into account the variability of the sea ice state. The interannual variability of sea ice drift speed based on buoy data can result from variability in the atmospheric forcing, oceanic forcing, and sea ice state as well as from the uneven spatial sampling of the buoy data. However, Rampal et al. (2009) and Gimbert et al. (2012b) investigated the trends in sea ice motion based on the IABP data, and ruled out the influence of spatial sampling of the buoys on their results.

Reduction of the error is also apparent in maps of the spatially distributed error. From the Polar Pathfinder free drift input to the $\alpha_w$ parameterization, the low-speed bias is reduced across the entire Arctic domain (Fig. 6e,f). Whereas Polar Pathfinder consistently underestimates sea ice drift speed, the spatial distribution of the bias in the $\alpha_w$ parameterization indicates an overestimation of ice drift speed north of the Canadian Archipelago and along coastlines, and an underestimation of drift speed along the Transpolar Drift (both averaging out when taking the pan-Arctic mean). Similarly, the spatial pattern of the root-mean-square error shows a net improvement in $\alpha_w$ over the Polar Pathfinder free drift input (Fig. 6i,j). However, we note that the spatial distribution of the explained variance ($R^2$) does not improve significantly (Fig. 6a,b), which can be related to both the Polar Pathfinder free drift input and the $\alpha_w$ parameterizations relying on constant transfer coefficients, and therefore ignoring underlying variations in the sea ice and ocean state.

Surprisingly, taking sea ice concentration into account to estimate the seasonally-varying transfer coefficient does not improve the agreement between drift estimates and observations, even when compared to a constant transfer coefficient (see Appendix). This dependency, which we presume exists, may not be picked up in our minimization procedure because most of the buoy data are located in high ice concentration areas (85% of the buoy data points are in sea ice of concentration >95%). Almost no data come from buoys located along the Eurasian coastline where free drift and a looser pack ice is present, particularly in the summer.

## 4.2 Spatially varying transfer coefficient

Whereas the previous section focused on free drift parameterizations that use a single transfer coefficient for the entire Arctic domain, in reality, this coefficient displays important spatial variability (Fig. 4a). Local values of $\alpha$ and $\theta$ are obtained using the same least-squares optimization method (Eq. 7a) on co-located pairs of IABP buoy data and ERA5 wind data constrained geographically within a 75 km x 75 km search window centred on each target grid cell. The $\alpha$ coefficients are in the 0-2.5 % range, in line with previous estimates of the scale and spatial variability of transfer coefficient based on buoy data and geostrophic winds (Thomas, 1999) or in-situ wind measurements (Heorton et al., 2019). The magnitude of $\alpha$ is lowest north of the Canadian Archipelago and along coastlines, and is maximal in the Beaufort Gyre, in the Transpolar Drift, and along

the East Greenland current. This spatial pattern agrees well with fields of $\alpha$ estimated by Kimura and Wakatsuchi (2000) and Maeda et al. (2020), who build on the ratio of satellite-derived ice drift speed to geostrophic wind speed. We note that the turning angles obtained from local fits do not show coherent spatial structures (results not shown).

The corresponding mean and standard deviation of the drag coefficient ratio ($C_a/C_w$, Fig. 4d) - the key parameter governing the magnitude of the sea ice drift (McPhee, 1980; Harder and Fischer, 1999), - are 0.19 and 0.11, respectively, for $\rho_a = 1.3$ kg/m$^3$ and $\rho_w = 1026$ kg/m$^3$ (Fig. 4d). These agree with typical values used in the Arctic sea ice modeling community and derived from observations (0.22, Hibler 1979; 0.19, Leppäranta 2011; 0.2-0.25, Lu et al. 2016; 0.5 Heorton et al. 2019).

We use these spatially varying (but constant in time) transfer coefficients and turning angles, in addition to the ocean currents estimates, as the basis for the free drift parameterization $\alpha_{ij}$. The root-mean-square error (mean bias error) for the norm of the velocity and the u- and v-components is reduced to 4.8 (-0.5), 4.4 (0) and 4.1 (0) cm/s, respectively. The explained variance is also improved compared to parameterizations based on a single transfer coefficient, with an $R^2$ of 0.59 for $\alpha_{ij}$. However, the root-mean-square (42°) and mean bias error (3°) does not improve despite the introduction of spatially varying turning angles. One key feature of the $\alpha_{ij}$ parameterization is its ability to partially retrieve ice stoppage events, which can be seen in the small $|U_i| \sim 0$ peak in the distribution of free drift estimate velocities (Fig. 3h). This feature is present in the buoy data, and the $\alpha_{ij}$ parameterization is the only parameterization that captures this, due to its reliance on local transfer coefficients which better represents instances of landfast ice along coastlines, or other regions characterized by very slow moving ice due to strong ice interactions. A locally optimized transfer coefficient captures some of the extremums of high and low sea ice drifting speed, which therefore contribute to reducing the general error. The seasonal cycle of $\alpha_{ij}$ does not differ a lot from the parameterizations that use a single transfer coefficient (Fig. 5); since the maps of $\alpha$ and $\theta$ are static in time, the seasonal cycle remains underrepresented. The long term trend is also misrepresented: whereas the time series of yearly-averaged buoy drift shows an acceleration of $+0.39$ cm/s decade$^{-1}$, the $\alpha_{ij}$ incorrectly yields a negative trend of $-0.33$ cm/s decade$^{-1}$ (Fig. 5b). Improvement can however be seen for the root-mean-square error during winter, where the $\alpha_{ij}$ parameterization performs a little bit better than the other parameterizations, which we can again attribute to a better representation of slow-drifting ice due to locally optimized transfer coefficients.

Considering spatial maps of the error metrics, the most striking feature of a free drift parameterization based on locally optimized transfer coefficients is apparent in the mean bias error field, which is minimized across the entire Arctic domain (Fig. 6g). However, the spatial distribution of the explained variance ($R^2$) and the root-mean-square error do not improve significantly compared to previous scenarios involving a single transfer coefficient: this is again explained by the transfer coefficients being invariant in time, and therefore missing on seasonality and long-term trends in the buoy observations.

### 4.3 State-dependent transfer coefficient

Next, we note the striking similarity between the spatial distribution of climatological PIOMAS sea ice thickness in the Arctic Ocean (consistent with Laxon et al. 2013) and the observed spatial distribution of the transfer coefficient (Fig. 4a,b). The results

from this simple analysis suggest a linear dependence of the transfer coefficient on sea ice thickness:

$$\alpha(h) = \alpha_h(1 - \beta_h h_i)e^{-i\theta_h}, \tag{8}$$

where $\alpha_h$ is the maximum value of the transfer coefficient for the free drift regime, $\beta_h$ is an ice-thickness ($h_i$) parameter modulating $\alpha(h)$, and $\theta_h$ is the wind-ice turning angle for the thickness-dependent parameterization. The error minimization routine yields an $\alpha_h$ coefficient of 2.0% and a $\beta$ coefficient of $0.17 m^{-1}$, which translates into a net transfer term that decreases linearly from 2% to 0% when sea ice reaches a thickness of 6.1 m. The negative dependence of $\alpha$ on $h_i$ indicates that ice thickness acts as a proxy for the internal ice stresses that are not represented in the free drift formulation. To first order, regions populated by thick ice are also experiencing more convergence and larger internal ice stresses, and the resulting ice motion is slower than a pure free drift solution. The inclusion of a dependence on ice thickness therefore allows for departures from the true free drift, via a scalable transfer coefficient. Note that this is different from parameterizations for the air-ice or ice-ocean form drag, which is a function of ice thickness due to higher surface or bottom roughness (e.g.: Tremblay and Mysak 1997).

Using the thickness-dependent transfer coefficient $\alpha(h)$, the root-mean-square error (mean bias error) for the norm of the velocity and for the u- and v-components are respectively 5.1 (-0.5), 4.6 (0) and 4.3 (0) cm/s. This represents an additional improvement on the root-mean-square error of compared to the parameterization with a single transfer coefficient $\alpha_w$ (5.4 cm/s); but the error is remains slightly larger than for the $\alpha_{ij}$ parameterization based on static maps of the transfer coefficient. The pan-Arctic, time-averaged mean bias error is unchanged. With respect to the Polar pathfinder free drift input, the thickness-dependent free drift yields a total reduction of the error and mean bias error of 35% (5.1 vs 7.9 cm/s) and 88% (-0.5 vs -4.0 cm/s). Using a similar linear free drift model built on monthly-fitted values of the transfer coefficient and turning angle over the 1983-1987 period, Thomas (1999) obtain a root-mean-square error of 5.2 cm/s for entire Arctic Ocean. The similarity of the best fit error from two different time periods causes us to speculate whether ca. 5 cm/s might constitute a lower limit to the accuracy of free drift estimates from linear minimization, with free parameters.

The benefits of a thickness-dependent parameterization $\alpha(h)$ emerge when we consider the seasonal cycle and interannual variability of sea ice drift speed (Fig. 5). All of the other free drift parameterizations follow the seasonality of the wind speed (Fig. 5a). Including a dependence on thickness improves the representation of the seasonal cycle of sea ice drift speed as observed in the buoy data, by better capturing the peak of ice drift speed in October and by shifting the drift speed minimum from July to May. Free drift estimates are most heavily used in the summer, when less satellite-derived ice motion estimates are available, which highlights the importance of including time-dependent sea ice state variables in the estimation scheme. The $\alpha(h)$ thickness-dependent parameterization also displays a lower mean bias error in any given season (Fig. 5c). Compared to the Polar Pathfinder free drift input, the thickness-dependent free drift consistently reduce the mean bias error by 80% and the root-mean-square error by 35% throughout the seasonal cycle. The relative error is lowest in the summer ($\sim$0.75) and peaks in the winter ($\sim$1). Highest relative errors are expected in the winter, since the ice speed is at a minimum, and the root-mean-square error is largest due to wintertime ice interactions not being explicitly represented by a free drift model, but only partially taken into account via the dependence on ice thickness.

The time series of annually averaged sea ice drift speed (calculated from the free drift estimates at buoy locations) also reveals that the thickness-dependence – $\alpha(h)$ – contributes to capturing long-term trends in sea ice drift speed (Fig. 5b). The constant transfer coefficient parameterization $\alpha_w$ results in virtually no trend on the ice drift speed (-0.09 cm/s decade$^{-1}$, not significant), so does the Polar Pathfinder free drift input $\alpha_p$ (-0.01 cm/s decade$^{-1}$, not significant). As discussed above, for the $\alpha_{ij}$ parameterization, the trend is even reversed (-0.33 cm/s decade$^{-1}$). The thickness dependent $\alpha(h)$ parameterization yields a trend (+0.45 cm/s decade$^{-1}$) that is much closer to the buoy observations (+0.39 cm/s decade$^{-1}$). Transfer coefficients constant in time result in an overestimation of the buoy drift speed in the earlier part of the record, and an underestimation it in the later part of the record. This indicates that the climatological thinning of sea ice is key in driving the long term trend in ice drift speed in the Arctic, supporting the findings of Rampal et al. (2009) and Kwok et al. (2013), who respectively use buoy and satellite-derived ice motion to attribute the trend in ice drift speed to the thinning of sea ice and to the loss of multiyear ice. These results also echo the work of Gimbert et al. (2012a, b), who associate an increase in the magnitude of inertial oscillation along ice trajectories over the 1979-2008 period with the mechanical weakening of the sea ice cover. In addition, the new free drift estimates capture a majority of the interannual variability of annually averaged ice drift speed ($R^2 = 0.77$), which is largely missed in the Polar Pathfinder free drift input $\alpha_p$ ($R^2 = 0.22$). An additional feature of the time series of the annual relative error is that all the parameterizations show a similar pattern: the relative error is lower in the earlier part of the record and then slowly increases after the 2000s. This could be linked to increasing drift speed, which is also generally prone to a larger absolute error (Fig. 5f,h). Nevertheless, the $\alpha(h)$ parameterization remains among the top drift parameterizations presented herein, displaying the lowest relative errors, on par with $\alpha_{ij}$, while having a better representation of long term trends in ice drift.

Another feature of the thickness-dependent parameterization $\alpha(h)$ is the higher explained variance, spatially, with respect to IABP buoy drifts (Fig. 6d). The explained variance is better than for any of the other parameterizations with transfer coefficients static in time, which highlights again the role of thickness as a time-dependent variable representing the mean sea ice state and its effect on dynamics. Regarding the mean bias error, the first order linear relationship between $\alpha$ and $h_i$ imposes a cap on the transfer coefficient, which results in a tendency to underestimate drift speed in regions where the drift is the fastest (and overestimate drift speed where it is the slowest), as shown in Fig. 6h. The spatial mean bias error is therefore higher than for the $\alpha_{ij}$, which represents a trade-off between mean bias and larger explained fraction of spatial variance and more accurate long term trends. The relative error (root-mean-square error divided by mean buoy drift speed) is highest (>1) along coastlines and in regions of thicker ice (north of the Canadian Archipelago), and progressively decreases towards the central Arctic Ocean (Fig. 6p). The relative error is minimal (~0.4) along the Transpolar Drift Stream and along the East Greenland Current, south of Fram Strait. This can be understood as an indicator of the quality of the thickness-dependent free drift estimates for representing different regions and ice motion regimes. Along the Transpolar Drift Stream, in the absence of confining pressure to the south, since Barents Sea is open, internal ice stresses are minimal and the free drift regime offers a good description of the motion field. On the other hand, in the Canadian Arctic Archipelago, or along the Alaskan coastlines or Eurasian coastlines, effects such as landfast ice and internal stresses under onshore winds limit free drift. Under these circumstances, Eq. 5 is a poor approximation, and the linear free drift estimates lose their fidelity. In addition, ice motion vectors from satellite imagery

are often unavailable due to land contamination of the pixel, which leads a number of data providers to not provide ice motion estimates within 25km of the coasts for the Polar Pathfinder merged dataset (Tschudi et al., 2019b). Finally, the representation of free drift in Equation 5 is isotropic, making it inadequate for the treatment of coastlines. Attempts have been made at constructing an anisotropic response using vector regressional analysis (Rabinovich et al., 2007) to better represent coastal interactions, but this is beyond the scope of the present study.

The $\alpha(h)$ parameterization does not improve the directional root-mean-square error (mean bias error), which is $42°$ ($3°$), the same as the constant $\alpha_w$ and $\alpha_{ij}$ parameterizations (Fig. 3l). This is expected, as the $\theta_h$ turning angle parameter itself is constant; however, we note that a simple linear dependence of the turning angle on ice thickness does not seem to improve the directional error (see Appendix A). Interestingly, Togunov et al. (2020) report similar numbers: compared to drifting telemetry collars in the Hudson Bay, the Polar Pathfinder ice motion (essentially based on free drift estimates in that region) yields an average drift direction error of $2.6° \pm 53.9°$. We also note a dependence of the directional error on the ice drift speed (Fig. 7). The drift direction is inaccurate for low values of buoy drift speed, but the spread of the error reduces and centers around zero with increasing sea ice drift speed. Low directional errors for higher ice drift speed is also reported in other studies comparing Polar Pathfinder merged ice motion vectors to drifting telemetry collars (Togunov et al., 2020) and comparing passive microwave sea ice drift to ice-tethered profilers (Hwang, 2013). This result is expected for two reasons. First, instances of low sea ice drift speed, when not driven by low winds, can be the result of important internal ice stresses. Since internal ice stresses are not considered in the free drift momentum balance, the direction of free drift estimates over regions subject to coastal effects or populated with thick, compact ice will be different from buoy observations. Second, nonlinear behavior becomes more important at low velocity. The model error for wind itself depends on the speed, and important errors can result from small directional errors at low wind speed (Stoffelen, 1998). In addition, the momentum balance for sea ice exhibits a stronger nonlinearity at very low ice drift speed, and the assumptions that were made initially when posing a linear free drift model do not hold either (Thorndike and Colony, 1982). Looking forward, the development of a parameterization for the wind turning angle should be considered for improving the accuracy of free drift ice motion estimates. Leppäranta 2011 (eq. 6.7b) do a formal derivation for wind-ice turning angle in a free drift regime, showing it to be a function of Coriolis forcing divided by the ocean drag. While this might help reduce the error, the introduction of the ice drift speed in the parameterization of the wind-ice turning angle removes the elegance of the simple linear free drift parameterization. Other interesting approaches include Hongwei et al. (2020) and Park and Stewart (2016) who observe a cubic polynomial relationship between the wind turning angle and 10m wind speed, based on ice-tethered buoys deployed in the central Arctic in 2012.

### 4.4 Surface ocean current estimates

The under-ice ocean current estimates obtained from the error minimization procedure (for the thickness-dependent free drift parameterization) capture the general features of the oceanic surface circulation in the Arctic, clearly showing the Beaufort Gyre and the Transpolar Drift Stream (Fig. 8a). The speed of surface currents is approximately 70% of the long-term, time-averaged sea ice drift speed, in general agreement with the assumption made by Thorndike and Colony (1982) who provided early estimates of the mean Arctic ocean circulation from ice drift velocity based on buoy data. The currents are slower north

of the Canadian Archipelago, and faster in the southern branch of the Beaufort Gyre and along the eastern coast of Greenland. The current speeds follow a Rayleigh distribution, with a mean of 3.5 cm/s and a standard deviation of 4.3 cm/s. These results are qualitatively similar to those of Kimura and Wakatsuchi (2000), based on different reference periods, who make current estimates by subtracting the time-averaged passive-microwave ice motion from time-averaged geostrophic winds.

We produce estimates of the surface currents before and after 2000 (Fig. 9a,b). We choose to split the data (spanning 1979-2019) around year 2000 in order to obtain two periods of roughly equal lengths; this choice is also consistent with breakpoint analyses that indicate structural shifts in sea ice area near the turn of the century (Goldstein et al., 2018). Although the number of observations is lower in the earlier part (n=130,146) than in the later part (n=327,769) of the record, they are sufficient to reconstruct climatological surface current estimates. Results indicate a distribution in sea ice drift speed that has a higher mean for the later part of the record (see Fig 9c), consistent with the positive trend in sea ice drift speed reported by Rampal et al. (2009). The associated increase in current speed is most apparent along the southern branch of the Beaufort Gyre, from the Alaskan coastlines to the center of the Chukchi Sea. A striking feature of the pre/post 2000s oceanic circulation is the expansion of a branch of the Beaufort Gyre over the Northwind Ridge and the Chukchi plateau. These findings are in line with Armitage et al. (2017) and Regan et al. (2019) who report an acceleration of the currents in the southwestern Beaufort Sea and a north-westward expansion of the Gyre, based on geostrophic currents derived from satellite radar altimetry. An intensification of the geostrophic currents in the periphery of the Gyre is also seen in hydrographic data obtained over the 2003-2011 period from ice-tethered profilers and shipboard surveys (McPhee, 2013). The expansion of the Beaufort Gyre over the Chukchi plateau is also found in models, and its dynamics are investigated in more details in Regan et al. (2020). These findings are coherent with a trend towards more open water in the summer season as well as faster ice drift due to thinner (Fig. 9e) and less compact ice, which in turn accelerates the ocean surface layer, while no large-scale acceleration of the wind speed is found (Fig. 9f). We also relate these results to the increase in freshwater content in the Beaufort Gyre, which trends and interannual variations are well documented by Proshutinsky et al. (2019). Comparing trends in the wind field curl and sea surface height in the Beaufort Sea (from which the doming of the gyre can be assessed, hence the accumulation in freshwater), Giles et al. (2012) conclude that the transfer of momentum from the atmosphere to the ocean increased at the turn of the 2000s. Similarly, McPhee (2013) use hydrographic data and note the increased doming of the gyre over the 2003-2011 period, marked by larger downward Ekman pumping at the center, and steeper horizontal density gradients and faster geostrophic currents on the periphery of the gyre. Our findings of faster currents in the Beaufort Gyre, coinciding with a reduction in ice thickness, support these conclusions and highlight the contribution of dynamic processes to the freshening of the gyre. The impacts of the declining ice draft to the freshwater cycle in the Beaufort Gyre are explored in more details in Krishfield et al. (2014). Based on mooring and satellite altimetry data, they note the decrease in multiyear ice and its replacement by thin ice or open water, which caused a reduction of the solid fresh water volume in the gyre, and an increased contribution of liquid water to the accumulation of freshwater. Krishfield et al. (2014) observe a stabilization of the freshwater content in the Beaufort Gyre in recent years. In a recent study, Armitage et al. (2020) argue that the stabilization of the freshwater content is due to an increase in eddy kinetic energy dissipation, attributed to a thinner, more mobile pack ice.

Additionally, we present estimates of the ocean currents $U_w$ for the high and low sea ice drift speed seasons (Fig. 10a,b), respectively defined as July to December (n=239,515), and January to June (n=218,400). We refer to these as the 'summer' and 'winter' seasons, from the sea ice drift point of view. The currents tend to be faster in the summer, primarily in the Southern branch of the Beaufort Gyre, consistent with a seasonal reduction in ice thickness (Fig. 10c,e). We also note that the surface currents in Fram Strait and east of Greenland appear to be slightly slower in the summer, potentially explained by the minimum in the wind speed seasonal cycle. Interestingly, the general winter/summer contrast in estimated ocean currents speed resembles the pre/post-2000s difference, which in both cases can be related to differences in ice thickness. The long term trend in sea ice thickness reduction affects the ice-ocean system in a way that is analogous to the seasonal cycle: the presence of perennial ice in the pre-2000s was favorable to drift conditions that were similar to the winter climatology, whereas the modern transition towards a seasonal pack ice constitutes an ice-ocean system that resembles the summer climatology. Gimbert et al. (2012b) draw similar conclusions by investigating the magnitude of inertial oscillations in relation with the sea ice mechanical response before and after the 2000s: "the mechanical behavior of winter sea ice in recent years is comparable to that of summer sea ice in previous years".

Our buoy-derived surface currents can be used as an independent source of comparison for different ocean state estimates from observations or reanalyses (Fig. 11). We calculate the climatology of buoy-derived ocean currents over the 2003-2014 period for different Arctic ocean state products: ECCO (Fukumori et al., 2019), GLORYS (Mercator Ocean, 2017), PIOMAS (Zhang and Rothrock, 2003) and the CPOM/DOT geostrophic currents derived from the dynamical ocean topography (Armitage et al., 2017). The climatologies from ECCO, GLORYS, PIOMAS and CPOM/DOT yield the expected features of surface Arctic oceanic circulation, with GLORYS and PIOMAS showing marginally faster current speed than the other products. Our estimates are generally in good agreement with the other ocean velocity products, the pan-Arctic root-mean-square difference (and relative difference, defined as root-mean-square difference divided by the mean of each product) between our climatology and ECCO, GLORYS, PIOMAS and CPOM/DOT are respectively 2.4, 2.9, 2.6 and 3.8 cm/s (0.82, 0.83, 0.76 and 1.30). For the central Arctic, the difference in current speed is only $\pm$ 1 cm/s. A common pattern emerges: our sea ice buoy-derived estimates tend to be slightly slower north of the Canadian Archipelago and along the Alaskan coast, and faster north of Fram Strait and along the East Greenland current. The low speed bias north of the Archipelago can be due to a misrepresentation of the internal ice stress, both in the models and in free drift estimates. We note that our method tends to overestimate fast ocean currents, and underestimate slow currents. The atmosphere-driven ice motion is generally overestimated in regions with little-to-no sea ice motion (Fig. 6h), which consequently drives the ocean current estimate down in the fitting algorithm's attempt to match the observations. On the other hand, our parameterization imposes a cap on the transfer coefficient $\alpha_h$; therefore, in regions where ice drifts the fastest, an underestimation of the ice drift speed can drive the associated current estimate upward. The largest differences are located in the East Greenland current where our estimates are over 5 cm/s faster than all of the other ocean velocity products. When comparing to PIOMAS specifically, we note faster currents in the southern Beaufort Sea, and slower currents in the interior of the Beaufort Gyre. A difference in the placement of the center of action of the Beaufort Gyre could reasonably explain this pattern. We also note two regions, along the coastlines of the Canadian Archipelago and along the Alaskan North Slope, where GLORYS is significantly faster than our estimates of the

currents. The work required to dissect these differences in ocean current speed, however this is beyond the scope of the present study. Nevertheless, when considering the larger picture, the good agreement between our estimates of ocean currents in the central Arctic and the observations-based CPOM/DOT data, and the general similarities between our climatology and ECCO, GLORYS, PIOMAS indicate that the reanalyses capture reasonably well the general ocean circulation in the central Arctic.

Returning to the free drift estimates, a caveat of our approach is the integration of a fixed ocean current climatology to calculate ice motion. Sea ice drift speed variations have a significant seasonal signature on top of a long-term trend (Fig. 5a,b); and due to a strong coupling of the ice-ocean system in the Arctic, these variations of ice drift speed also drive variations of the surface currents. As explored by Meneghello et al. (2018) in model studies, the slowdown of the ice speed in the winter can lead to an inversion of the surface ice-ocean stresses; modulating the surface currents seasonally. With respect to the long-term trends, Armitage et al. (2017) report increased oceanic surface current speeds over the 2003-2014 period, concurrent with a thinner, weaker and looser sea ice cover. The inclusion of these two features of the oceanic currents variability in our free drift parameterization would help to further improve the representation of the seasonal cycle and the long-term trends in ice drift speed. Advancing the representation of the coupling between sea ice and the ocean, Park and Stewart (2016) develop an analytical free drift model that includes an ice-ocean boundary layer with an Ekman spiral.

## 5 Conclusions

Wind-driven ice motion estimates are an essential component of merged ice motion datasets (e.g.: Polar Pathfinder, Tschudi et al., 2019b), providing information on sea ice drift when neither buoy observations nor satellite-derived drift vectors are available. In this study, we present new estimates of sea ice motion based on the free drift of sea ice, introducing a sea ice state dependent wind-ice-ocean transfer coefficient. Free drift is defined from the balance of the surface wind stress and ocean drag; i.e. excluding internal ice stresses, Coriolis and sea surface tilt. This results in the standard linear relationship: $\boldsymbol{U}_i = \alpha \boldsymbol{U}_a + \boldsymbol{U}_w$ (Thorndike and Colony, 1982; Leppäranta, 2011), where $\boldsymbol{U}_i$ is sea ice velocity, $\boldsymbol{U}_a$ is wind velocity, $\boldsymbol{U}_w$ is the surface ocean current, and $\alpha$ is the transfer coefficient. Using a 40-year record (1979-2019) of buoy data from the International Arctic Buoy Program and 10m wind velocity from the ERA5 reanalysis, we find a spatial co-variability between the transfer coefficient $\alpha$ and sea ice thickness ($h_i$) from the PIOMAS reanalysis (Zhang and Rothrock, 2003). The transfer coefficient is lowest in regions populated with thick ice (e.g.: north of the Canadian Arctic Archipelago) and increases in the peripheral Arctic seas which are seasonally covered with much thinner sea ice. The spatial pattern of the transfer coefficient is consistent with Maeda et al. (2020), increasing our confidence in the results. Based on these results, we parameterize the transfer coefficient as a linear function of sea ice thickness such that: $\alpha(h) = \alpha_h(1 - \beta_h h_i)e^{-i\theta_h}$, where $\alpha_h$ and $\beta_h$ are free parameters controlling the amplitude of the transfer coefficient, and $\theta_h$ is a wind turning angle. We use a least-squares minimization approach and obtain the following values for the free parameters: $\alpha_h = 2.0\%$; $\beta = 0.17 \text{ m}^{-1}$ and $\theta_h = 25°$.

We use the thickness-dependent free drift parameterization $\alpha(h)$ to produce ice motion estimates, which we compare to the 40-year record of buoy drift. The proposed parameterization relies on sea ice thickness from the PIOMAS reanalysis that is produced in near-real time. The free drift estimates yield root-mean-square errors (mean bias error) of 5.1 cm/s (-0.5 cm/s) for

the drift speed and 42° (3°) for the drift direction. This represents a 35% reduction of the error on the ice drift speed compared to the free drift estimates presently used in the Polar Pathfinder dataset. Although a parameterization ($\alpha_{ij}$) that builds on static maps of the transfer coefficient and the turning angle yields the lowest time averaged pan-Arctic root-mean-square error and the lowest mean bias error (spatially), key advantages of the thickness-dependent free drift $\alpha(h)$ are better representations of the seasonal cycle, the interannual variability, and long-term trends of sea ice drift speed. Only the thickness-dependent parameterization reproduces a long-term trend in sea ice drift that aligns with the observations. Including a dependence on ice thickness further shifts the seasonal cycle of ice motion estimates towards observations. The seasonal cycle of the error is characterized by a minimum relative error in the summer period (July-October), which is a desirable feature since free drift estimates are mostly used in the summer season when a lesser number of ice motion vectors from satellites are available.

In the minimization procedure, the ocean currents appear as the part of the signal in sea ice drift speed that is not explained by surface winds (Thorndike and Colony, 1982). The climatology of these surface ocean current estimates captures the general features of the Arctic Ocean general circulation, including the Beaufort Gyre, the Transpolar Drift Stream, and a fast outflow current along the eastern coast of Greenland. We explore climatological changes in the ocean state by retrieving the ocean currents for the pre- and post-2000 periods. We observe a general acceleration of the surface currents, concurrent with a reduction in sea ice thickness from the pre-2000s to the post-2000s. The most striking feature of the climatological changes in the ocean state is a sharp acceleration of the southern Branch of the Beaufort Gyre and an expansion of the Gyre over the Chukchi Rise, a feature also reported in other studies using different methods (McPhee, 2013; Armitage et al., 2017). Our approach provides independent estimates of the wind- and ice-driven component of the surface ocean currents; they can be used in conjunction with observations of the currents from altimetry or ice-tethered profilers for improving our understanding of the Arctic Ocean circulation.

This simple, observations-based, optimized and cost-effective model aims at providing ice motion estimates for the length of the observational record. Extensions to this work could investigate the balance between complexity and accuracy, when introducing increasingly complex processes, from linear free drift models all the way up to models with a complete description of sea ice physics. Future work includes the development of an optimal interpolation approach for integrating our free drift estimates to other ice motion inputs provided as part of the Polar Pathfinder dataset; and performing an intercomparison of different Arctic ocean state estimates and our independent surface ocean current climatology.

*Code and data availability.* Code and data are available at https://web.meteo.mcgill.ca/~charles/freedrift/

**Appendix A: Other formulations**

A number of different, equally intuitive, parameterizations were also tested, yet they did not reduce the root-mean-square error with respect to the buoy data (Table A1). We report on them here to inform others interested in similar questions. In fact, the parameterization based on ice thickness alone, or static maps of the transfer coefficient and turning angle, are the simplest and

**Table A1.** List of additional free drift parameterizations tested in the context of this study. The error metrics are based on a comparison to the IABP buoy sea ice drift data over 1979-2019.

| | | Coefficients | | | | Error metrics | | | | |
| | | | | | | RMSE and (Mean bias error) | | | | |
| Name | Parameterization | $\alpha$ [%] | $\beta$ | C | $\theta$ [°] | $\|U_i\|$ [cm/s] | $u_i$ [cm/s] | $v_i$ [cm/s] | $\theta$ [°] | $R^2$ |
|---|---|---|---|---|---|---|---|---|---|---|
| $\alpha(A)$ | Eq. A1 | 1.6 | 1.2 | 35.6 | 24 | **5.5** (-0.5) | **4.9** (0.0) | **4.4** (0.0) | **43** (3) | 0.49 |
| $\alpha(A,h)$ | Eq. A2 | 2.0 | 1.7 | 0.41 | 24 | **5.2** (-0.5) | **4.7** (0.0) | **4.3** (0.0) | **43** (3) | 0.54 |
| $\theta(h)$ | Eq. A3 | 1.3 | -2 | - | 28 | **5.4** (-0.4) | **4.9** (-0.1) | **4.5** (-0.1) | **42** (9) | 0.48 |

provide the lowest root-mean-square error. Following Hibler 1979, we first tested a wind-ice-ocean transfer coefficient with that decays exponentially with sea ice concentration:

$$\alpha(A) = \alpha_A - (\alpha_A - \beta_A)e^{-C(1-A)}. \tag{A1}$$

The three free parameters $\alpha_A$, $\beta_A$ and $C$ represent the free drift transfer coefficient, the fully ice-covered transfer coefficient, and a decay coefficient between free drift and full ice cover. Secondly, we tested a transfer coefficient that has a dependence on both sea ice concentration and thickness:

$$\alpha(A,h) = \alpha_{Ah} - (\alpha_{Ah} - \beta_{Ah})h_i e^{-C(1-A)}, \tag{A2}$$

where the free parameters have the same meaning as for the previous equation. Lastly, we tested a thickness-dependent wind turning angle, taking into account that the deviation from the surface wind will be larger for thicker ice because of the linear dependence of the Coriolis parameter on ice thickness:

$$\alpha_\theta e^{-i(\theta_h + \beta_h h_i)} \tag{A3}$$

None of the above formulations lead to an improved parameterization of the drift velocity and angle. The root-mean-square error on the $\alpha(A,h)$ parameterization is similar to that of the $\alpha(h)$ parameterization based on ice thickness alone, indicating that the predictive skill all lies in the knowledge of the thickness field.

*Author contributions.* CB and BT designed the study and prepared the manuscript. RN and BT conducted preliminary data collection and analysis. CB and BT conducted the present version of the study. CB retrieved the data, developed the code and prepared the figures. All authors collaborated in editing the final manuscript.

*Competing interests.* No competing interests are present.

*Acknowledgements.* This research was supported by the Forecasting Regional Arctic sea ice from a Month to Seasons (FRAMS) project, which is funded by the Marine Environmental Observation, Prediction and Response Network (MEOPAR). This research received additional funding from the Office of Naval Research (ONR-N00014-11-1-0977), the National Science Foundation (NSF-PLR 15-04404), the Canadian Sea Ice and Snow Evolution Network (CanSISE), which is funded by the Natural Science and Engineering and Research Council (NSERC) Climate Change and Atmospheric Research Program. Charles Brunette is grateful for academic and financial support by ArcTrain Canada, McGill University, Québec-Océan, Fonds de recherche du Québec - Nature et technologies and the Norht Slope Borough of Alaska (Eben Hopson Award). The authors wish to thank Laure Coquart for retrieving and providing the ERA5 data, and Rym Msadek and Gunnar Spreen for their input at different stages of the project. Maps were created using the ncpolarm function created by Andrew Roberts (2020), and the Scatter Plot coloured by Kernel Density Estimate function by Nils Haëntjens (2020). Directional statistics were computed using the Circular Statistics Toolbox for Matlab by Philipp Berens (2012). The authors wish to acknowledge the following data providers: the National Snow and Ice Data Centre, the International Arctic Buoy Programme, the Copernicus Climate Change Service, the Polar Science Centre (University of Washington), the Centre for Polar Observation and Modelling (University College London), the Jet Propulsion Laboratory (NASA), and Mercator Océan. The authors thank Thomas Lavergne, Harry Heorton and an anonymous reviewer, and editor Michel Tsamados for their constructive comments that helped improve the quality of the manuscript.

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

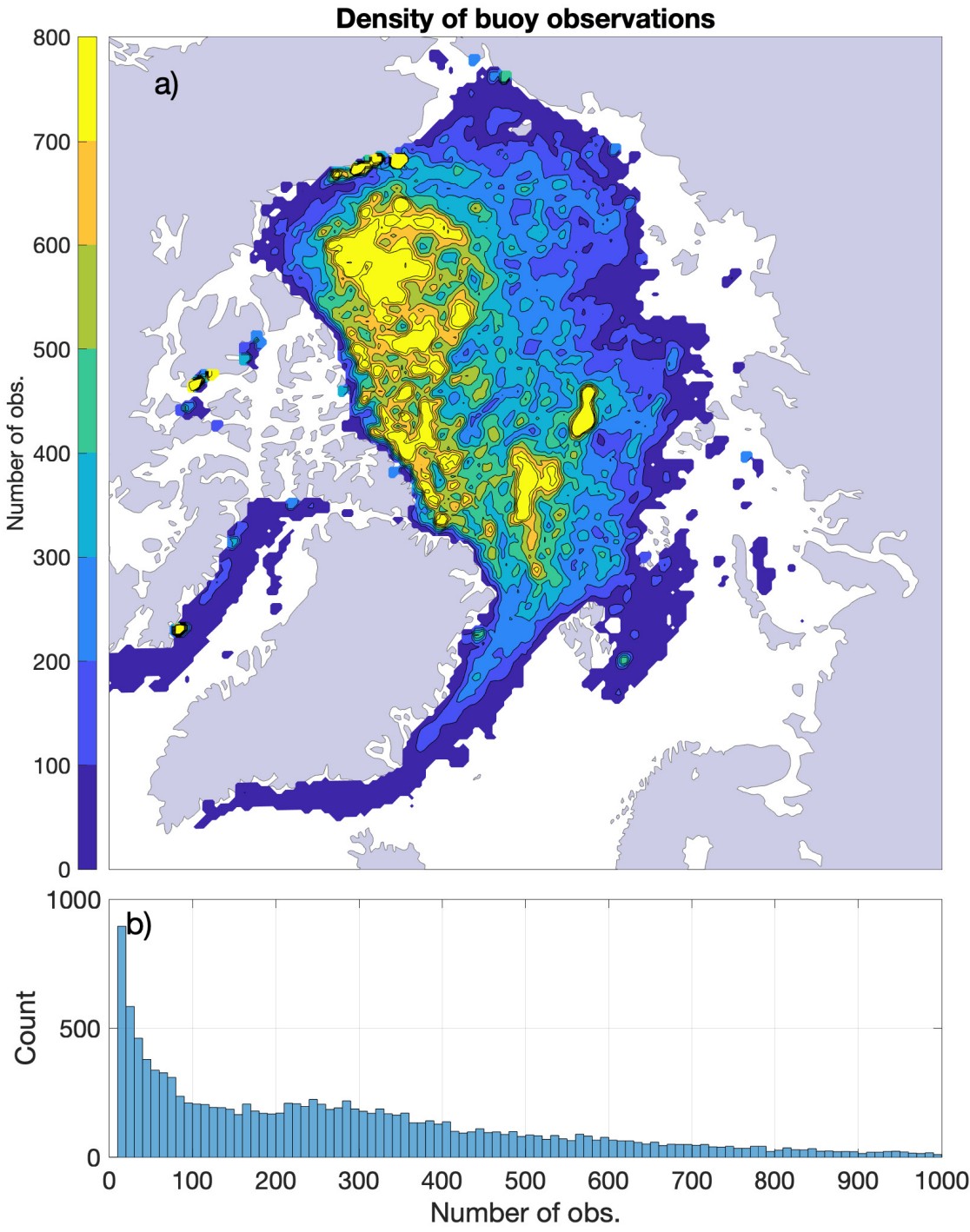

**Figure 1.** a) Based on the density of IABP buoy drift data over the full 1979-2019 period: spatial map of the number of data points used for the evaluation of the surface currents $U_w$ (and $\alpha$ when applicable) at every grid location. b) Distribution of the number of points in grid cells.

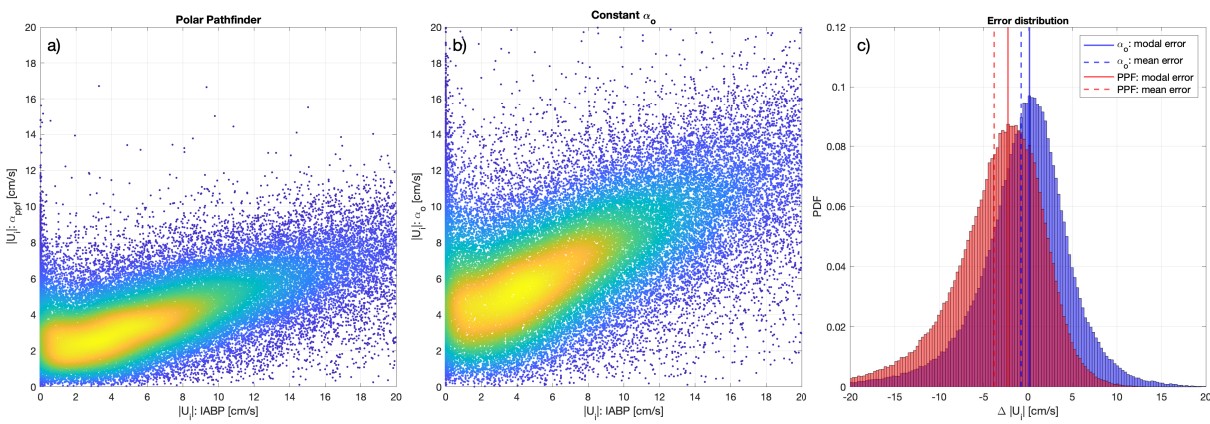

**Figure 2.** Scatter plot of free drift sea ice speed estimates against buoy ice drift speed for: a) the NSIDC Polar Pathfinder free drift input data, and b) for linear free drift with an optimized wind-ice-ocean transfer coefficient (no ocean currents). The color scale indicates the density of points on the scatter plot (kernel density estimate). The distribution of the error with respect to the buoy data is shown in c), solid lines indicate the modal bin; dashed lines indicate the mean error.

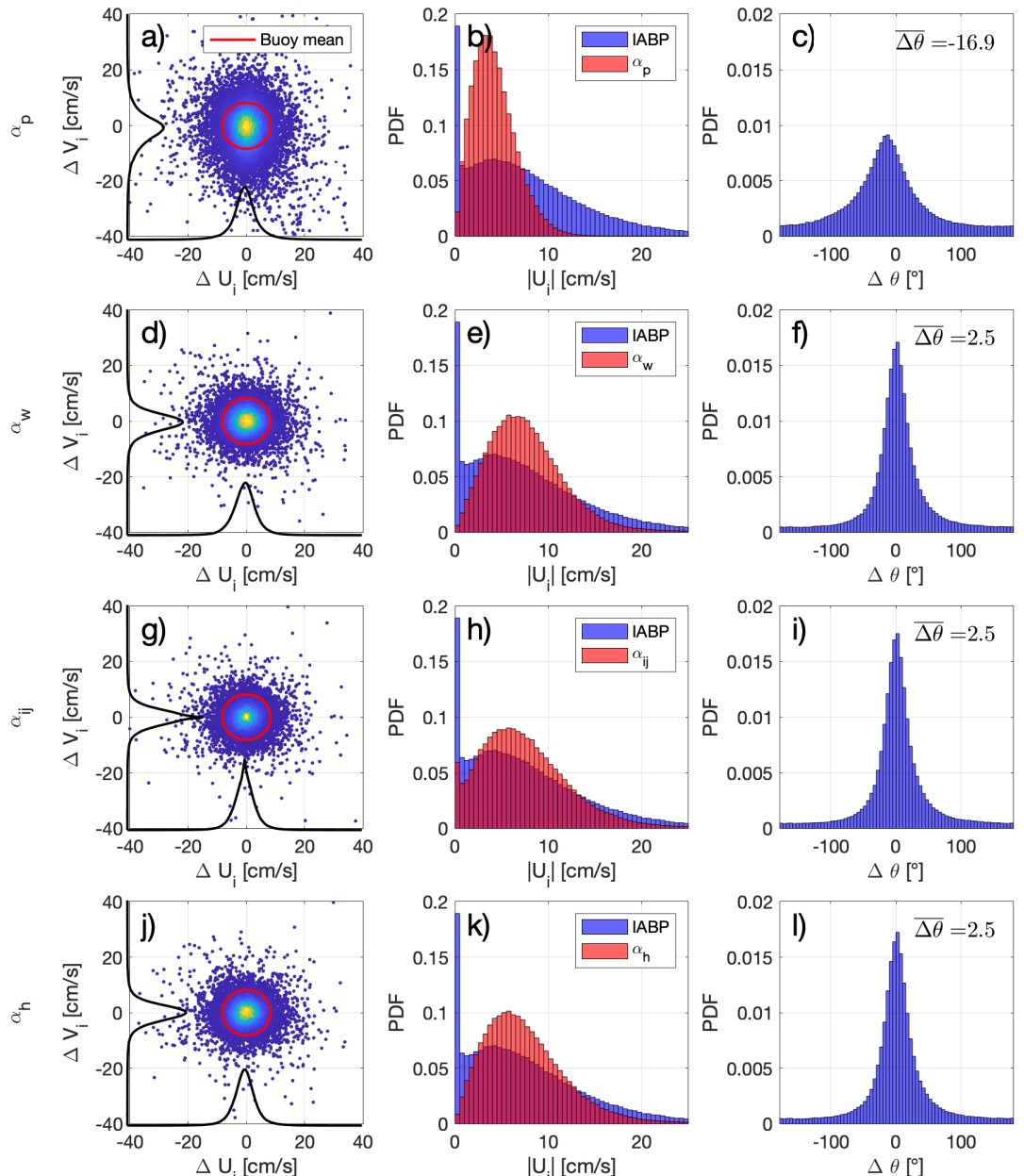

**Figure 3.** Error metrics for the the $\alpha_p$ Polar Pathfinder free drift input (a-b-c), constant $\alpha_w$ free drift with ocean currents (d-e-f), spatially varying transfer coefficient and turning angle (g-h-i) and $\alpha_h$ thickness dependent free drift parameterization (j-k-l), taking the IABP buoys as the reference. Left column: scatter plots of the error on the u- and v-components of sea ice velocity, where the color scale indicates the density of points, the red circle has a radius equal to the mean buoy drift speed, and the black curves illustrates the distribution of the error in each direction. Middle column: distribution of the sea ice drift speed for the buoy data (blue) and for each of the free drift estimates (red). Right column: distribution of the drift direction error. All available data over the 1979-2019 period is included.

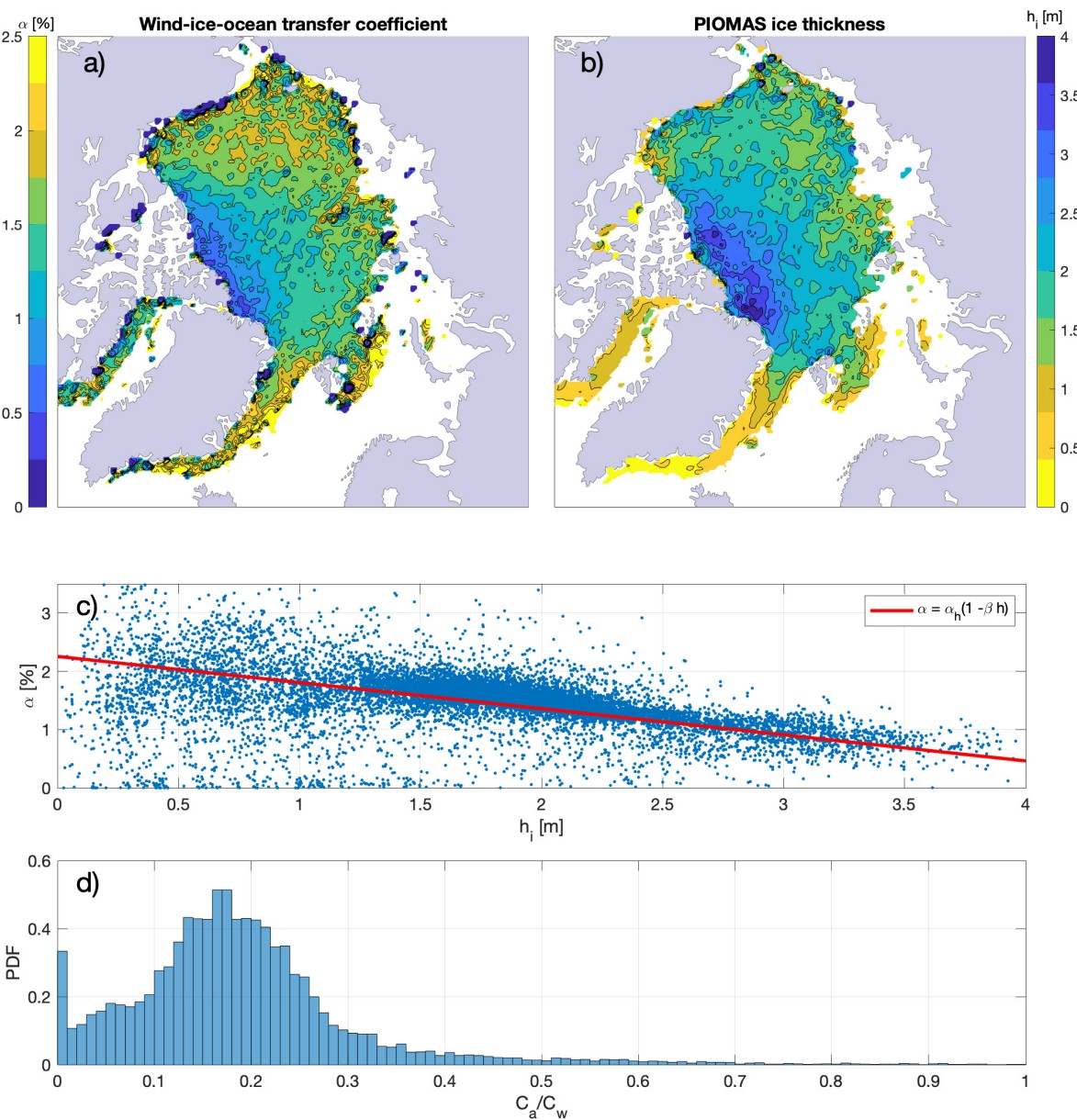

**Figure 4.** Spatial distribution of: a) wind-ice-ocean transfer coefficient calculated from a local linear fit between the IABP buoy ice velocity and the ERA5 10m wind velocity, b) PIOMAS ice thickness (at buoy location). For each grid cell, all data over the 1979-2019 period within a 3x3 search array centered on each grid cell is considered. These fields represent year-round climatological averages. c) scatter plot of the magnitude of the transfer coefficient against ice thickness; the red line is the best linear fit. d) distribution of the equivalent ratio of atmospheric to ocean drag coefficients, $C_a/C_w$.

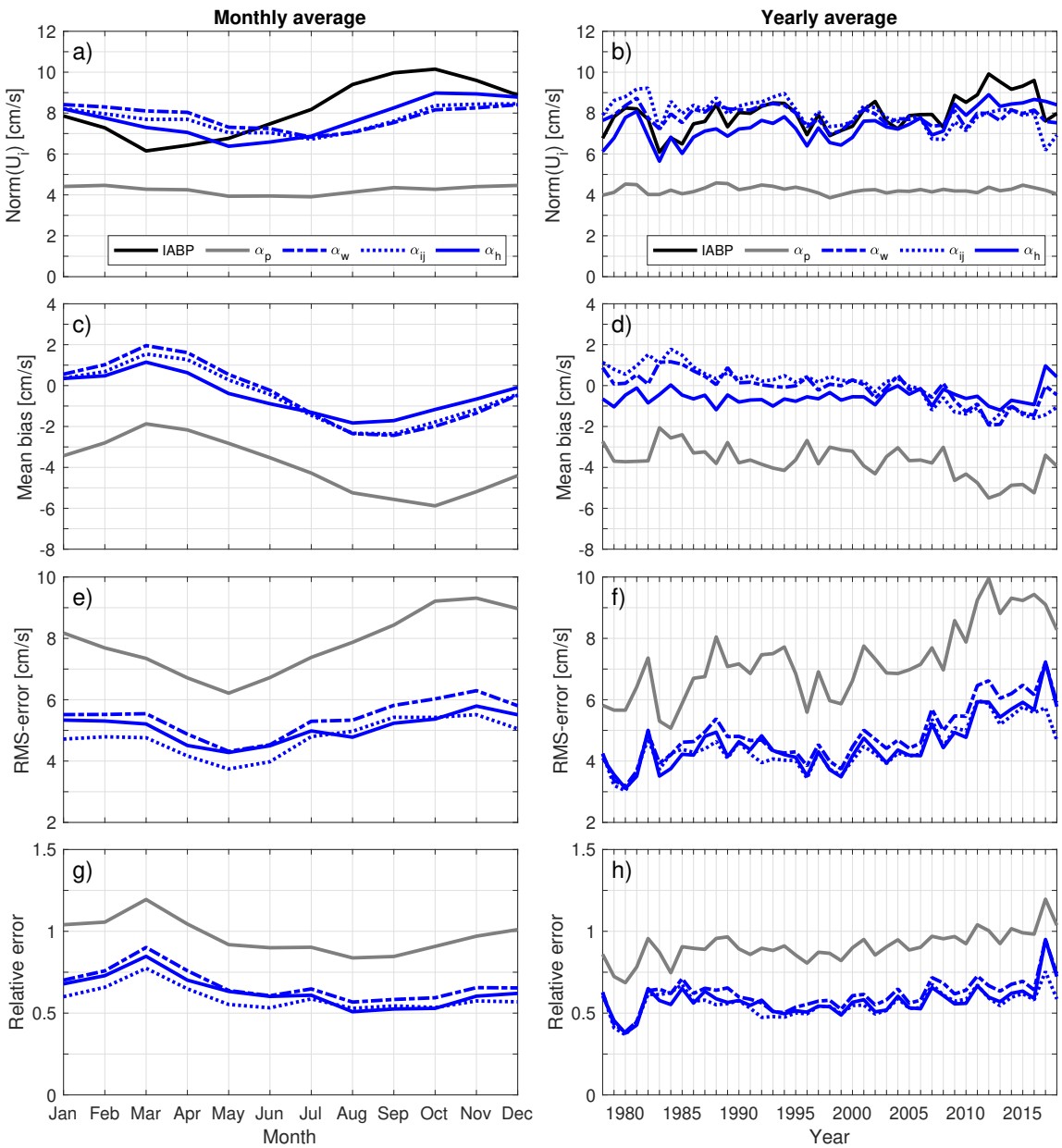

**Figure 5.** Comparison between co-located IABP buoy drifts and the Polar Pathfinder free drift input ($\alpha_p$, grey), free drift based on a single wind-ice-ocean transfer coefficient ($\alpha_w$, dashed blue), free drift based on static maps of the transfer coefficient and turning angles ($\alpha_{ij}$, dotted blue), and the free drift including a dependence on sea ice thickness ($\alpha_h$, blue). All three $\alpha_w$, $\alpha_{ij}$, and $\alpha_h$ parameterizations take into account ocean current estimates. Plots show the seasonal cycle of monthly averaged: a) drift speed c) mean bias error, e) root-mean-square error, g) relative error; and the 1979-2019 yearly averaged time series of b) drift speed d) mean bias error, g) root-mean-square error, h) relative error. The relative error is defined as the root-mean-square error divided by the mean buoy drift speed.

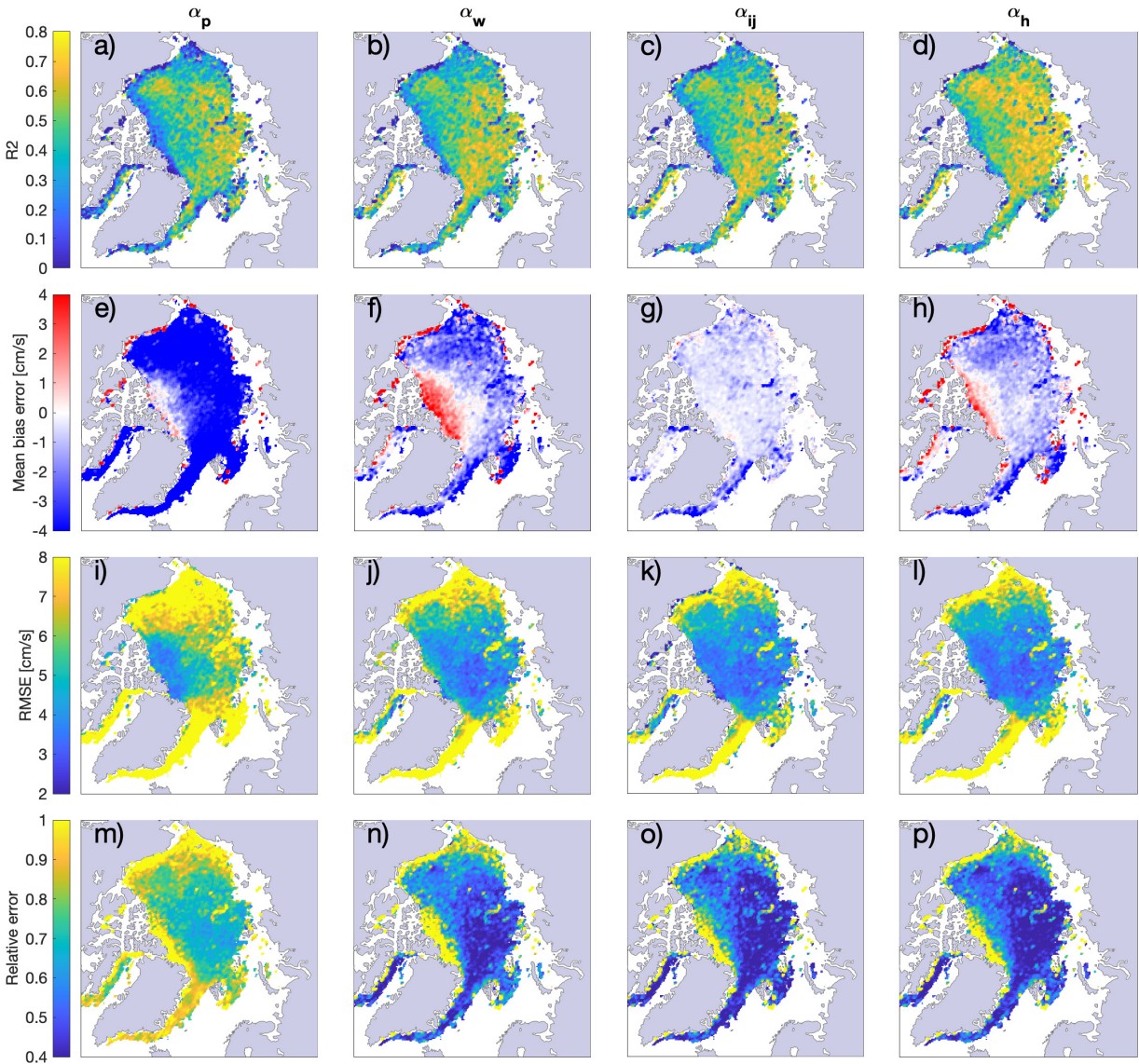

**Figure 6.** Spatial maps of error metrics comparing IABP buoy drift to: Polar Pathfinder free drift input ($\alpha_p$, first column), free drift based on a single wind-ice-ocean transfer coefficient ($\alpha_w$, second column), free drift based on static maps of the transfer coefficient and turning angles ($\alpha_{ij}$, third column), and the free drift including a dependence on sea ice thickness ($\alpha_h$, fourth column). Error metrics include: the explained variance ($R^2$, first row), the mean bias error (second row), the root-mean-square error (third row), and the relative error, defined as the root-mean-square error divided by the buoy mean (fourth row). All data points over the 1979-2019 period are included.

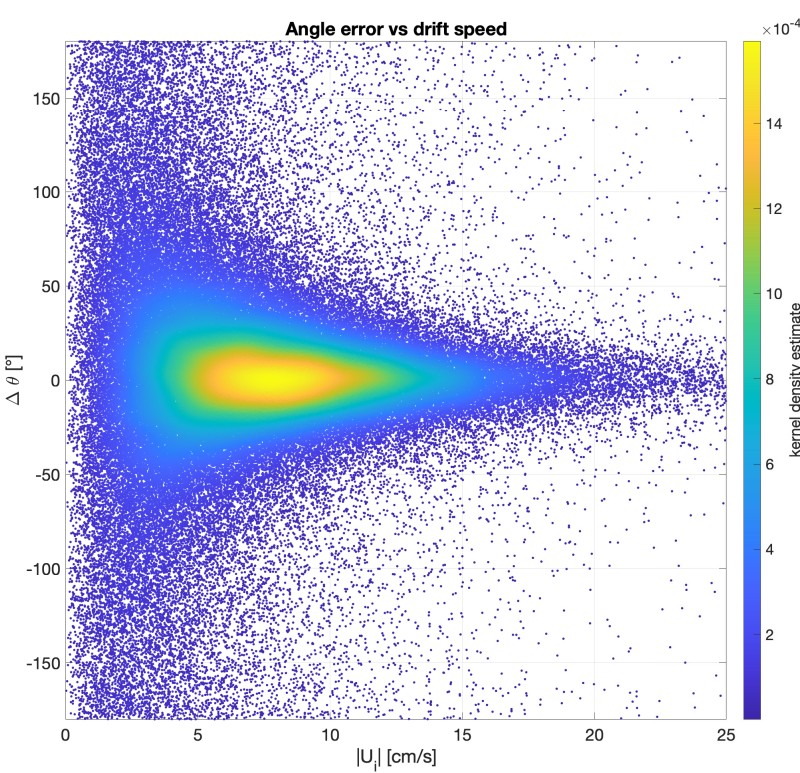

**Figure 7.** Error on the direction of the drift as a function of the IABP buoy drift speed, for the thickness-dependent parameterization $\alpha_h$. The error is taken as the angle between the direction of the free drift estimate and the direction of the buoy drift, for all data available over the 1979-2019 period. The drift angle error takes a value between +180° (free drift is to the left of the buoy drift) and -180° (free drift is to the right of the buoy drift). The color scale represents the kernel density estimate, an indicator of the density of points in the graph.

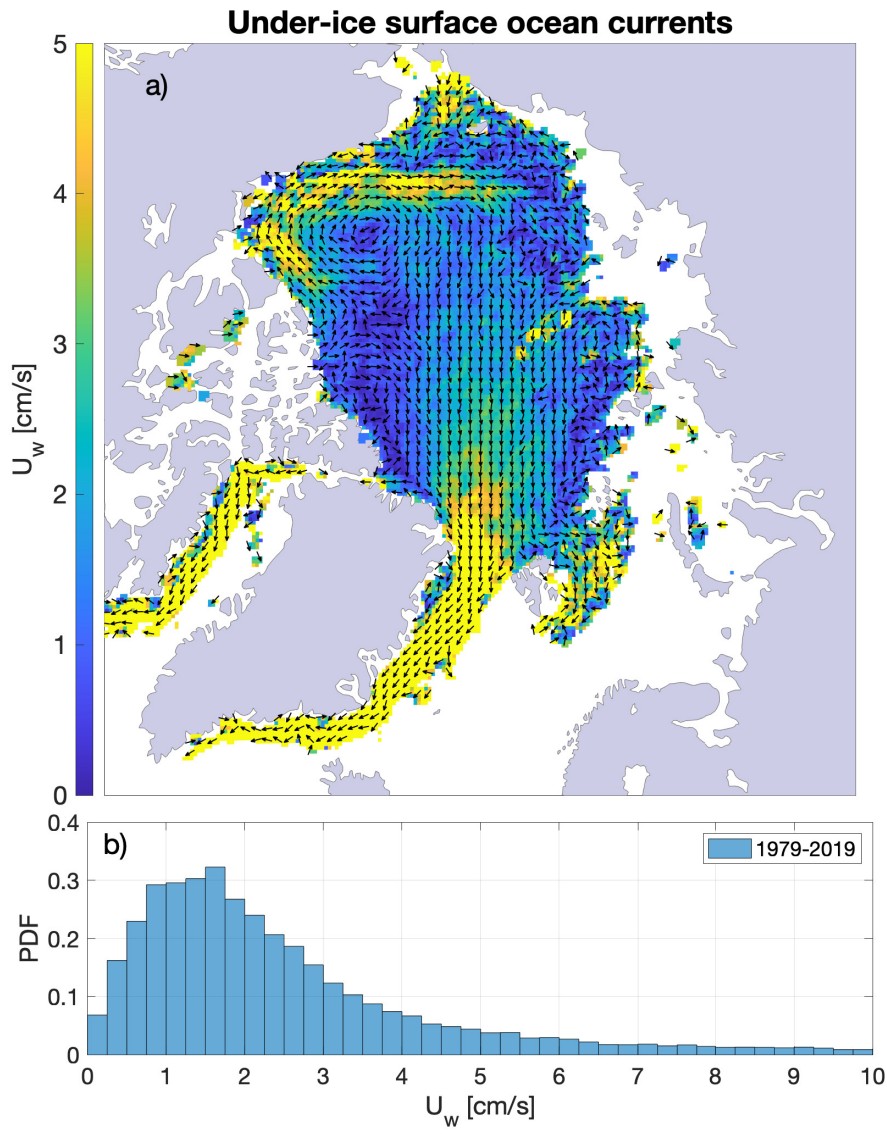

**Figure 8.** a) Estimates of the oceanic surface currents over the 1979-2019 period; calculated as the ice motion unexplained by the wind, considering the $\alpha(h)$ parameterization for free drift. The distribution of the current speed is presented in b). For any given grid cell, the surface current is optimized as an intercept in a best linear fit between the IABP buoys sea ice drift and the ERA5 10m winds.

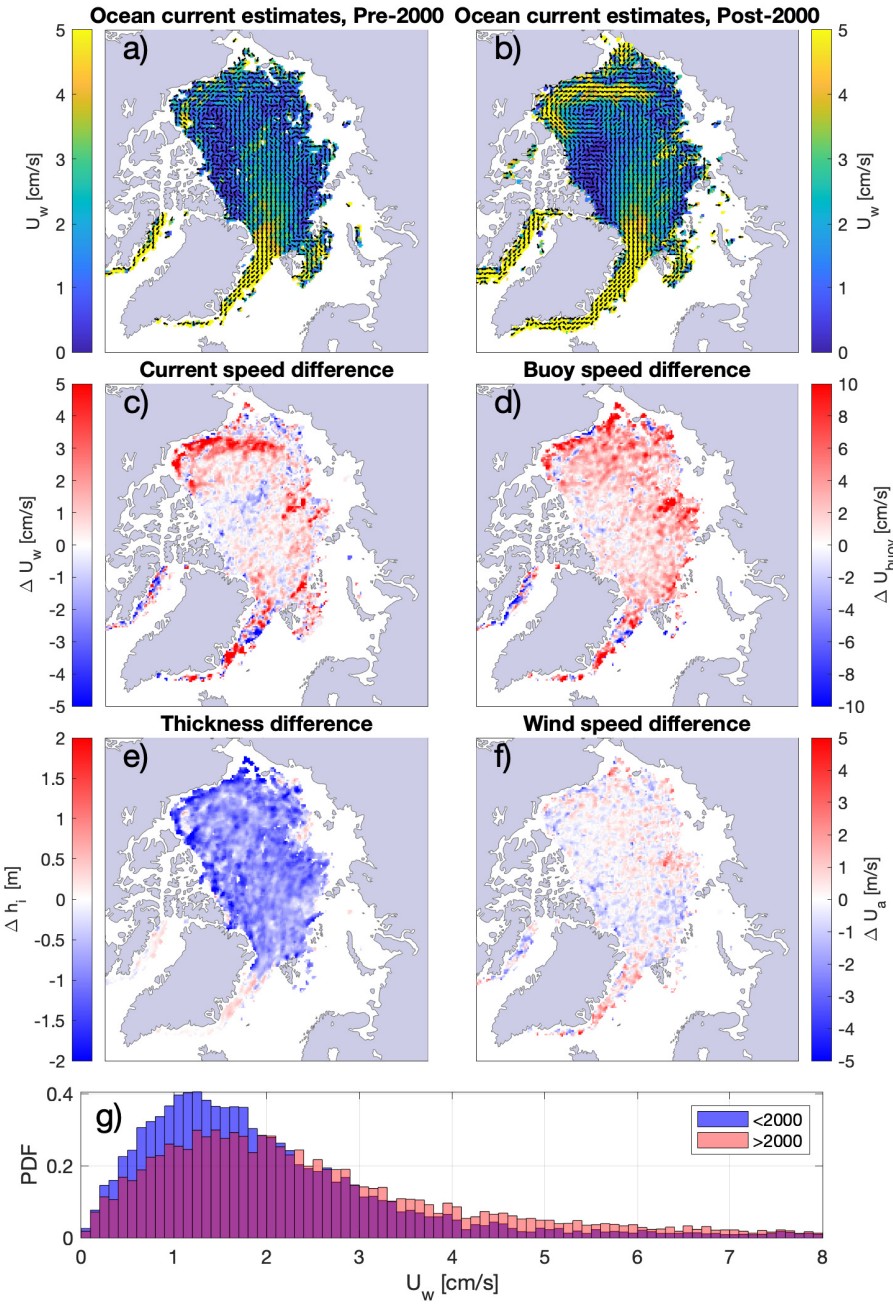

**Figure 9.** Estimates of the oceanic surface currents for the period covering a) 1979-2000, b) 2001-2019. Maps of the difference between the later and earlier part of the record (post-2000 minus pre-2000) are presented for the following fields: c) surface oceanic currents speed, d) IABP buoy-derived ice drift speed, e) ice thickness from PIOMAS, and f) 10m wind speed from ERA5. The distribution of current speed for the pre- and post-2000 periods is presented in g).

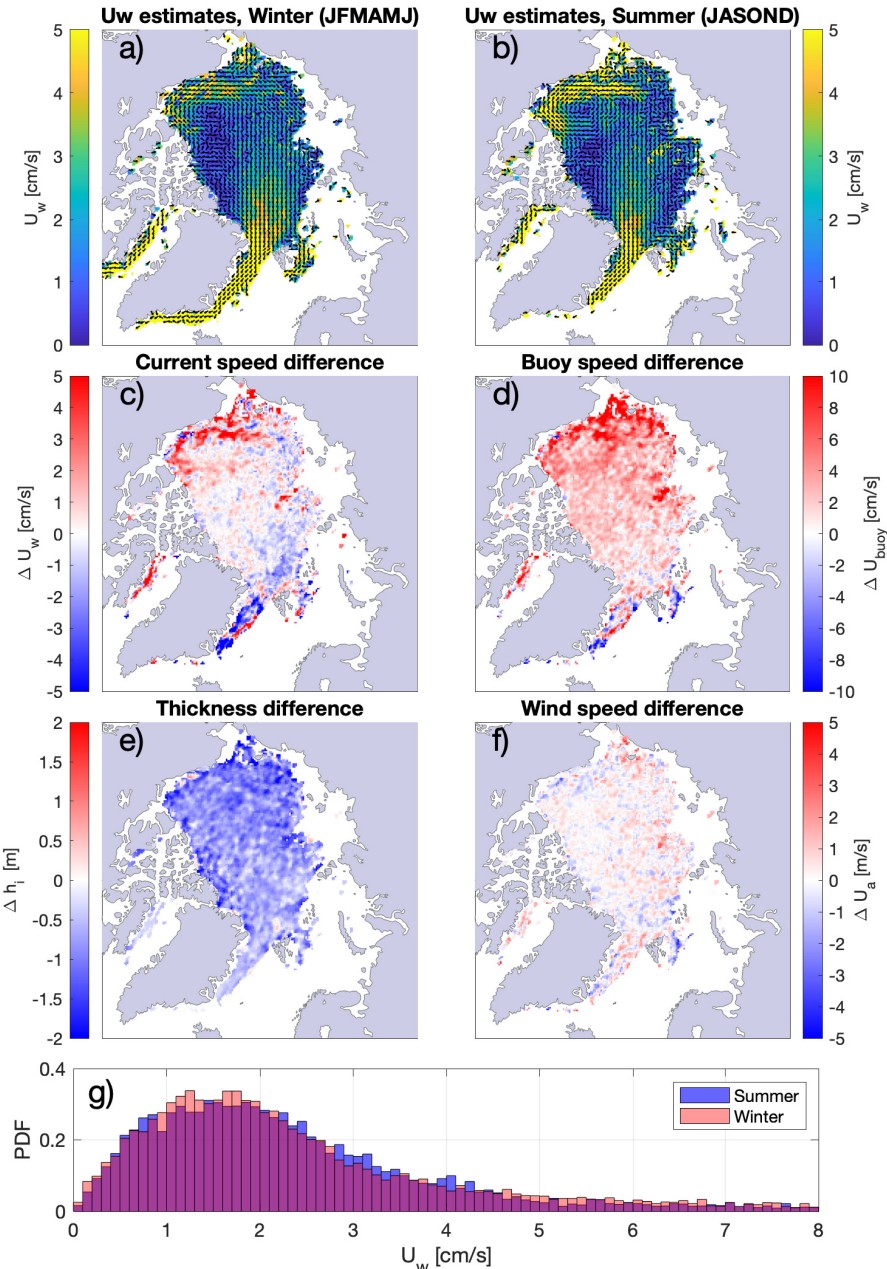

**Figure 10.** Estimates of the oceanic surface currents for a) the summer season (JASOND) b) the winter season (JFMAMJ). Maps of the difference (summer minus winter) are presented for the following fields: c) surface oceanic currents speed, d) buoy-derived ice drift speed, e) ice thickness from PIOMAS, and f) 10m wind speed from ERA5. The distribution of current speed for the summer/winter seasons is presented in g).

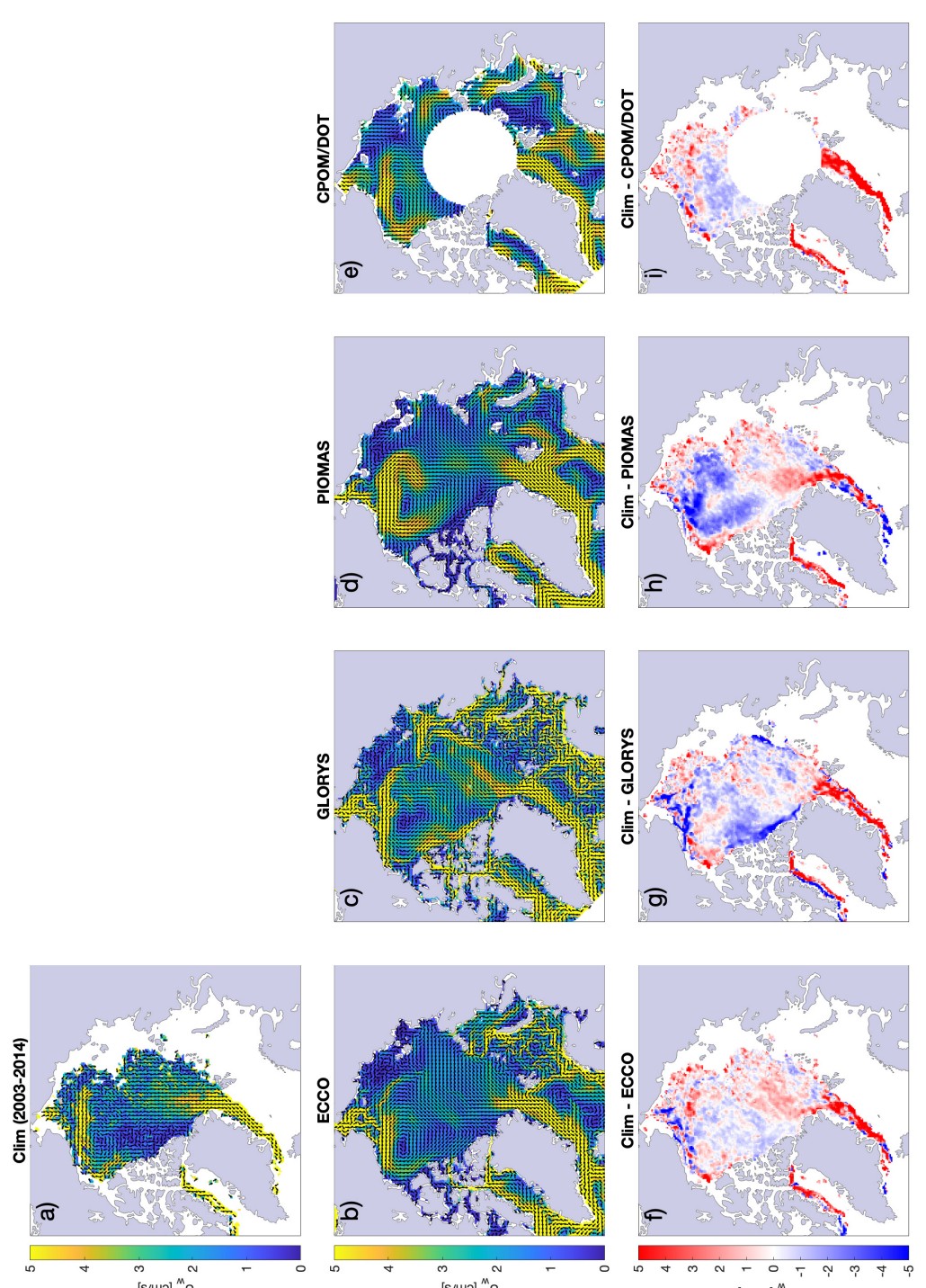

**Figure 11.** 2003–2014 climatology of the ocean currents from: a) intercept in the linear free drift parameterization between IABP buoy data and ERA5 winds, b) ECCO, at a depth of 5 m c) GLORYS, at a depth of 5 m d) PIOMAS, at a depth of 7.5 m and e) geostrophic ocean currents from the dynamic ocean topography. f) through i) shows the difference in velocity between the ocean currents in a) and each of the four other ocean velocity product.