# Peer review of "A new state dependent parameterization for the free drift of sea ice"

_The Cryosphere, 2021_

## Referee Comment (RC1)

Review of **tc-2021-249:** *A new sea ice state dependent parametrization for the free drift of sea ice*, by Brunette et al.

The authors present a new parametrization of the free-drift model for sea-ice motion. Their new model considers a linear parametrization of the wind-ice transfer coefficient on sea-ice thickness. The new model is compared to earlier versions of the free-drift model including that currently implemented in the NSIDC PathFinder V4 sea-ice drift dataset. The resulting motion fields are analyzed in terms of RMSE against buoy data (themselves used for training), seasonal cycle, and multi-decadal trends. The ocean currents (obtained as residuals during the tuning of the free-drift model) are compared to other currents from ocean/ice reanalyses and satellite altimetry data. The new currents confirm independently several recent findings from other investigators including the freshening and widening of the Beaufort Gyre post-2000, providing further confidence in the new approach.

The paper is sound and well written. The methodology and data are clearly explained, and the analysis of the results in terms of ocean currents in the Beaufort Gyre brings interesting additional knowledge from a paper that could have been "just" a method paper. The paper is very relevant for EGU TC and can be published with minor edits.

I would nevertheless invite the authors to give some thoughts about the following, and amaybe reflect some of them towards the end of the ppaer: a major element for the attractiveness of the free-drift models is simplicity. The model is simple to tune, simple to implement, simple to run. Only the wind field (for tuning and running) and sea-ice motion data are needed (for tuning). The new state-dependent free-drift model is not as simple since it requires space/time varying sea-ice thickness as input. Today, such daily complete fields come mostly from complex ocean/ice models, that have required the assimilation of satellite products, tuning, forcing etc… With such a machinery underpinning the results, one wonders e.g. if the new free-drift model performs much better than the PIOMAS sea-ice velocities (that are as accessible as the thicknesses, and do include a rheology). Did you compare your free-drift velocities to PIOMAS' and can you prove the free-drift model adds information?

By the same token, the seasonal cycle obtained from the new parametrization is very convincing. But would it have been present if the free-drift model had been parametrized on a sea-ice thickness climatology (a "simpler" concept)? In the same vein, would the approach of Thomas (1999) (seasonally varying air drag) provide a good-enough seasonal cycle without the need for any sea-ice thickness information?

In short, I invite the authors to reflect on the trade-off between the improved performance of their free-drift model vs the added complexity it brings, noting that PIOMAS' velocities (or from alternative ocean/ice models) are as available as their sea-ice thicknesses.

**Other comments:**

It seems the importance of sea-ice motion tracking is described in two blocks, first L39-44, then L116-123. Maybe combine these two blocks (or make it clearer why they stand apart). In these justifications: the Lagrangian tracking of sea-ice age is a key use for year-round sea-ice drift data.

L130-135: What is the justification for not using EASE2? If the justification is to stay close to the NSIDC PathFinder grid, please mention it.

L165: Andersen et al. 2007 did not look at the NSIDC CDR (it had not been released). Kern et al. 2019 and Kern et al. 2020 do evaluate the NSIDC CDR V3 (among 9 other CDRs).

Kern, S., Lavergne, T., Notz, D., Pedersen, L. T., Tonboe, R. T., Saldo, R., and Sørensen, A. M.: Satellite passive microwave sea-ice concentration data set intercomparison: closed ice and ship-based observations, The Cryosphere, 13, 3261–3307, https://doi.org/10.5194/tc-13-3261-2019, 2019.

Kern, S., Lavergne, T., Notz, D., Pedersen, L. T., and Tonboe, R.: Satellite passive microwave sea-ice concentration data set inter-comparison for Arctic summer conditions, The Cryosphere, 14, 2469–2493, https://doi.org/10.5194/tc-14-2469-2020, 2020.

Section 3.2: The methodology section was not very clear and did not contain all the methodologies involved. I suggest to:
- Move what currently is at the start of section 4 (L246-256) into section 3;
- Also include the parametrisation on sea-ice thickness in section.
- Move L224: "Note that ..." to when the 1$^{st}$ velocity statistics are commented in section 4.
- Clarify what you mean with L226-230. Is this really needed for the iterations to converge (in my experience with the free-drift model, it converges fast without those steps).

Section 4: Methodology of the validation: did you use a circular-statistics version of the RMSE for the angles? If yes, specify it. If not justify why not.

L266: "are further reduced"? Here we are in a chain of reductions expressed as %age and it should be clear if the reference is the initial state, or the previous step.

L271-273: this sentence about ERA-Interim results was not very clear, and interrupts the flow of ERA5-based results. Maybe re-formulate and move elsewhere?

L312: "and the net wind-ice ..." would it be easier to state the value of Beta? The reader excepts Beta but it is not directly given.

L322-326: what seasonal cycle did Thomas (1999) achieved with his monthly tuning? If this is not known, can we hypothesize?

L370-371: ice-ice interactions are a possible explanation. But also the functional relationship from dX,dY to direction is highly non-linear for small velocities, hence small errors in dX,dY will bring high errors in the direction. Maybe refer to Stoffelen 1998 (Appendix B) for these non-linear relationships and their impact on error propagation?

Stoffelen, A. (1998), Towards the true near-surface wind speed: Error modeling and calibration using triple collocation, *J. Geophys. Res.*, **103**, 7755–7766.

L401: Please add a reference for the pre/post 2000 breakpoint in the ice trend. This break is not obvious from the sea-ice extent data.

L445-458: Here you discuss the differences between your currents and those from the other sources. You bring twice that the sea-ice rheology in the modelling framework might be too approximated, leading to errors in their currents. However, your own estimates neglect ice-ice interactions altogether. How can you be sure that the issue is on the ocean/ice modelling side and not on your side? Maybe revise to bring more balance in this discussion.

L487: This is maybe where one could balance the improvements brought by the new methods with its added complexity.

Appendix A: it is very commendable that you took the efforts to document what did not work very well.

Acknowledgments: Maybe add acknowledgments for the data providers (C3S, NSIDC, IABP, etc...)

**Editorials:**

Title: you have "sea-ice" twice. Maybe drop the 1st of them (A new state dependent…)?

L3: "Building on the fact" → "Observing"

L4: "a structure as the" → "a structure similar to that of"

L13 : "mean and root-mean square error": it took me a while to decipher. Maybe "mean error and root-mean square error" is more readable?

L20: what do you mean by "observations acceleration"? The observed acceleration?

L53: Lavergne et al. **2010**. Check the publication year.

L154: Consider cite https://rmets.onlinelibrary.wiley.com/doi/full/10.1002/qj.3803 for C3S/ECMWF ERA5 data.

L161: Given the recent release of their V4, you could specify that you used V3 of the NOAA/NSIDC SIC CDR.

L364: "use" is maybe not the right word since your ocean currents are a result?

L364-391: this discusses the limitations that your ocean currents are fixed over time. But we haven't seen the ocean currents yet. Consider moving this text towards the end of the ocean-current section.

L425: missing space between "the" and "gyre"

L427: "this stabilization" (was described in the previous sentence).

L437-438: analogous to the seasonal cycle ("to" missing). Similar to… ("to" missing).

L467-469: "This established … coefficient": this sentence seems broken.

L489: missing opening parentheses for the citation to Thorndike and Colony

Fig 10, panel g): the transparency of red/blue did not allow to fully observe the two distributions.

---

## Referee Comment (RC2)

This paper documents the matching of wind driven free drifting sea ice to observations of ice tethered buoy drift. A number of methodological scenarios are considered.
First a constant parameter set for the equations of freed drift are sought.
Secondly the ocean surface currents are considered with climatological maps of ocean currents retrieved along with the free drift parameters.
Finally the free drift parameters are replaced with sea ice thickness dependent parameters.
I suggest this paper for publication. The writing is technically accurate with few typos and grammatical errors. The results shown are compelling and make a useful contribution to sea ice science. In particular it is very interesting to see the wind driven and ice thickness dependent components of sea ice drift quantified. Also the retrieval of ice surface currents and their comparison to other ocean surface data and models is very interesting.

However, I found reviewing the paper quite frustrating and challenging. A few kept parts of the method are lacking, which makes the interpretation of the complex results rather difficult. In particular the description of the Pathfinder data used, and the description of the 'Polar Pathfinder' parameterisation makes many parts of the paper difficult to interpret. The Pathfinder data is described in the data section as pathfinder free drift data, that from the description seems to be a custom made free drift data set using a NCEP/NCAR reanalysis, whilst descriptions of the data in the paper appears to be the version 4 pathfinder dataset I am familiar with, that includes many floe tracking and ice buoy additions to the free drift equations. What data is used here? I suggest finding an alternate description of the base 'control' free drift parameterisation (using the same parameters used to create the Pathfinder dataset) within the paper in order to make it much easier to read.

The description of the methods used in section 3.2 requires more information. The paper appears to have three key scenarios where various parameters are retrieved and additional data is also retrieved (ocean currents, wind-ice transfer coefficient in figure 4(a)?). It will be much easier to understand these methods if each can have an equation (7) describing exactly how the values are retrieved. Also in this section more information is required on which data are collected for each case. It is confusing at times how many buoy observations, how many reanalysis based free drift equations and how many parameter values are considered in each scenario. For example are you seeking maps of \alpha and \theta or single values? The case of allowing for variable ocean surface currents is very difficult to understand. Do you actively search of ocean currents with each pixel a free parameter? Or is it simply the residual in the equation as mentioned later in the text.

I suggest the paper for publication after minor revisions. While the paper was difficult to interpret at times, the quality of the figures and results suggest that it will al work out if the points above are addressed.

H. Heorton.

Specific points.

L4-5 'has a structure as the spatial distribution of sea ice thickness'
Please reword this sentence as the above is very difficult to interpret
L 5 please include to what you have introduced this parameterisation. Which model are you using?
L 13 'minimize'  - minimizes
L13 what cost function?

It is unclear from the abstract what exactly you have undertaken in this study.

L 25 I enjoyed this quotation. Thank you

Section 2.3

Here you describe a Pathfinder Data set I am unfamiliar with. The citation and DOI given point me towards the Polar Pathfinder version 4 data set. This data is the ice motion data set derived from

Floe tracking algorithms and the IABP buoy data set you mention above along with NCEP/NCAR reanalysis. Is the data you are using?

The free drift data you mention in this section appears to be a conversion from NCEP/NCAR surface winds through free drift ice drift equations only (using constant parameters). This data is then interpolated onto the 25km EASE grid. Is this correct?

Did you create this data yourself? If so can you title this section appropriately. For instance if this free drift data is NCEP/NCAR derived free drifting sea ice estimates, then please call it this.

Section 3.2

Equation (7). If I have interpreted the text from line 225 correctly, then $U_i^{fd}$ in equation (7) is also a function of $U_w$. Is this correct? If so is it possible to show this within equation (7)? As in $U_i^{fd}(\alpha,\theta,U_w)$? If you use different equations at different times, then perhaps listing all the equations used will aid the reader.

Whilst this section technically well written I find it hard to follow the exact methodology. Evaluation of equation (7) at a single data point is under constrained, so I assume there is a high number of n data points. You mention that $U_w$ is only varying spatially. What temporal and spatial resolution do you use for the IABP data, the ERA5 data and for the searched for $\alpha$ and $\theta$? What is a typical value of n? How do you co-locate the temporarily varying IABP data with the static but spatially varying $U_w$, and how does this relate to the extracted parameterisations? The results and conclusions suggest that you create spatially varying maps of $\alpha$ and $\theta$, and then the results suggest you only extract single values for each case, can you clarify? For discussion of the role of time averaging of drift and wind data in the free drift calculations see Heorton et al. 2019.

Section 4,

I'm struggling to understand what has been performed during each scenario. What data is used in which part of the equation and what is being solved for in each case?

L 240 So does fig 1 indicate the density of IABP buoy drift data?

Table 1, can you expand on what exactly is meant by the mean bias error metrics. Each case has two numbers, which is which? How is the value in brackets for ui and vi all equal to 0.0 for every case except for $\sigma_p$. What is this number? Is it the mean dimensional bias for this value? Does this show that the error for this value is evenly spaced around the mean?

L 252, Polar Pathfinder parameterisation. This line is difficult to interpret when the form of this data is unclear. If the data here is free drift ice equations applied to NCAR/NCEP are you extracting $\alpha$ and $\theta$ by comparing winds from ERA5 to free drifting sea ice created using NCAR/NCEP? Do you replace the IABP buoy data in equation (7) with the free drift sea ice data set? It is unclear in section 3 and equation (7) how you evaluate buoy data, free drifting sea ice gridded data, and the minimisation of two free parameters at each point, or for the full data set. Or do these cases just imply that you have applied some parameters to colocated IABP drift and ERA5 winds, and then compared this to the free drifting sea ice calculated from NCAR/NCEP winds (the Polar Pathfinder data)? If this is the case, then this is a strange comparison of two different surface wind reanalysis data sets. If this Polar Pathfinder dataset is the floe tracking derived ice drift data from the citation in section 2.3, and not a 'free drift' data set created for this study, then this comparison makes sense.

L 253, does constant wind-ice transfer coefficient mean single values of $\alpha$ and $\theta$ were calculated for all the data? Or do you produce static maps of $\alpha$, $\theta$?

L 260 Can you comment more of the differences for the $\sigma_p$ case? Here you are comparing this difference between 'Polar Pathfinder' parameterisation, a new constant parameterisation and the IABP buoy drift vectors? Is this correct? Is this Pathfinder the v4 data set? If this is the case

then the changes discussed in this section come from either the parameters in the free drifting ice used in the Pathfinder dataset, the correction to this from floe tracking and buoy drift performed as part of the creation of the Pathfinder v4 dataset, or the use of NCAR/NCEP compared to the ERA5 dataset used in this paper? If however this 'Pathfinder free drift dataset' is of your own creation, then what comes from using a different reanalysis and what comes from changing the parameters?

L 261 In this case ($\sgima_0$) is a single parameter extracted for $\alpha$ $\theta$ for all data?

L 265 Is this case not $\sigma_w$? See also figure 3.

L 280 this peak in zero-velocity ice drift suggests that this is the floe tracking Pathfinder drift data set. This confusion in what Pathfinder data you are comparing to is making this interpretation of your results very difficult.

L299 please describe how the value of wind-ice transfer coefficient is calculated here. Is this a map of extracted values of $\alpha$? Or the ratio $|U_i|/|U_w|$? Do you compare the directional velocity in this ratio, or simply the drift speed? An equation here may be helpful.

L 302 Also of comparison here is Heorton, H.D.B.S. et al. 2019.

L 323 bias-corrected parameterisation is $\sigma_0$? Please check all text descriptions of $\sigma_w$ and $\sigma_0$. Are you including the consideration of ocean currents in figure 5?

L 326-328 Is this sentence referring to the Pathfinder drift data you are comparing to? If this is the case then this is the v4 data including the floe tracking and buoy data.

L 327 are you using time-dependant variables here, or is this a suggestion for future work?

L 425 'thegyre'

L 428 Please reword the last part of this sentence, it is difficult to interpret

L 430 What are the buoy data counts for these two cases as you give on line 402 for pre/post 2000?

L 451 Here you point out the limitations of your model in that it does not resolve sea ice rheology. I think it is also worthwhile to discuss the physical limitations in your representation of the ocean surface currents. You previously cite the Meneghello et al. (2018) paper where the inversion of sea ice to ocean stress is described. Do you think your method of resolving ocean currents using wind forced sea ice free drift may only resolve the wind driven (through the sea ice cover) component of ocean surface currents? Can you account for cases where the ocean currents drive ice drift during low wind speeds? What further information may be presented by including geostrophic currents and resolving Ekman currents?

L 459 I think it is important to point out that your ocean surface currents do not resolve the northern, eastward part of the Beaufort Gyre.

L 470 It is unclear from the methodology where and when and how you created the maps of wind-transfer coefficient that you compare to the sea ice thickness data.

L 488, how do they appear as a residual? Equation 7 and the associated text refer to them as a value to be solved for.

L 489 please correct the parentheses.

L 493 - 'post-200s' to 'post-2000s'

L 496 It will be more accurate here to describe them as the 'wind driven component of ocean surface currents'

Figure 3 caption. Here you list the columns as (pathfinder $\sigma_p$, free drift $\sigma_0$ and thickness dependant $\sigma_h$) but the figure labels show the middle column as $\sigma_w$ for ocean currents. Please re-write accurately.

Figure 4, which case is this map of $\alpha$ calculated for?

Figure 5, It is unclear which data is compared here. Is this the averaged drift only where there is co-located IABP and parameterised ice drift data? If not then this is an unfair comparison. How do you calculated the relative error? Is it part of the use of equation (7)? Also again I'm unsure what the Pathfinder drift data is. There are large differences in these plots between the parameterised ice drifts and the buoy data. Visually your new free drift parameterisations are much closer to the IABP drift data than the 'Pathfinder' parameterisation, whilst the relative error plots show only a slight reduction, can you explain this? What time periods do the data represent? The ERA5 winds used here are daily averages. Are the buoy drifts also daily averages?

Figure 7, which case does this figure relate to? Please indicate this in the caption.

Figure 9, can you clarify the definition of the difference plots? Is it (post 2000) - (pre 2000)?

References:
Heorton, H.D.B.S. et al. 2019. Retrieving Sea Ice Drag Coefficients and Turning Angles From In Situ and Satellite Observations Using an Inverse Modeling Framework. Journal of Geophysical Research: Oceans. 124, 8 (2019), 6388–6413. DOI:https://doi.org/10.1029/2018JC014881.

---

## Author Response (AR1)

**TC-2021-249: Response to Reviewers**

**RC1: 'Comment on tc-2021-249', Thomas Lavergne, 30 Aug 2021**

The authors present a new parametrization of the free-drift model for sea-ice motion. Their new model considers a linear parametrization of the wind-ice transfer coefficient on sea-ice thickness. The new model is compared to earlier versions of the free-drift model including that currently implemented in the NSIDC PathFinder V4 sea-ice drift dataset. The resulting motion fields are analyzed in terms of RMSE against buoy data (themselves used for training), seasonal cycle, and multi-decadal trends. The ocean currents (obtained as residuals during the tuning of the free-drift model) are compared to other currents from ocean/ice reanalyses and satellite altimetry data. The new currents confirm independently several recent findings from other investigators including the freshening and widening of the Beaufort Gyre post-2000, providing further confidence in the new approach.

The paper is sound and well written. The methodology and data are clearly explained, and the analysis of the results in terms of ocean currents in the Beaufort Gyre brings interesting additional knowledge from a paper that could have been "just" a method paper. The paper is very relevant for EGU TC and can be published with minor edits. *We thank the referee for this positive evaluation of the manuscript!*

I would nevertheless invite the authors to give some thoughts about the following, and maybe reflect some of them towards the end of the paper: a major element for the attractiveness of the free- drift models is simplicity. The model is simple to tune, simple to implement, simple to run. Only the wind field (for tuning and running) and sea-ice motion data are needed (for tuning). The new state- dependent free-drift model is not as simple since it requires space/time varying sea-ice thickness as input. Today, such daily complete fields come mostly from complex ocean/ice models, that have required the assimilation of satellite products, tuning, forcing etc... With such a machinery underpinning the results, one wonders e.g. if the new free-drift model performs much better than the PIOMAS sea-ice velocities (that are as accessible as the thicknesses, and do include a rheology). Did you compare your free-drift velocities to PIOMAS' and can you prove the free-drift model adds information?

The work presented here addresses the free-drift component of an observations-based sea ice drift data set that covers the full seasonal cycle from 1979 to today (i.e. the Polar Pathfinder). We are currently working on a revised, biased-corrected, optimally interpolated (using weights that are function of the spatially varying decorrelation scale) version of Polar Pathfinder with complete spatial and temporal error characterized, that is based on buoy data, satellite-derived sea ice drift and the free drift estimate presented here. The comparison proposed by the reviewer is important and will be done in that follow-up project, when we can compare the full seasonal cycle of observational (revised Polar Pathfinder) and simulated (PIOMAS, including SIC assimilation) data set (estimated submission date: Fall 2022).

In its current state, the thickness-dependent free drift product relies on an ice thickness reanalysis product (PIOMAS), which reproduces the observed cycles and trends in sea ice volume within reason; our objective is to produce a free drift ice motion dataset that covers the length of the observational record. As sea ice thickness observations will become more readily available in the future, new ice thickness data streams would in practice replace the use of reanalysis thickness. The present study paves the way towards these fully observations-based free drift estimates.

By the same token, the seasonal cycle obtained from the new parametrization is very convincing. But would it have been present if the free-drift model had been parametrized on a sea-ice thickness climatology (a "simpler" concept)? In the same vein, would the approach of Thomas (1999) (seasonally varying air drag) provide a good-enough seasonal cycle without the need for any sea-ice thickness information?

A seasonally-varying air drag, as in Thomas et al. 1999, would contribute to an improved representation of the seasonal cycle, but not to a reduction of errors spatially; and while a climatology of ice thickness would reduce the spatial error to some extent, it would not take into account the seasonal change nor the decadal changes associated with the warming of the Arctic. The thickness-dependent parameterization captures much better the long-term positive trend in ice drift speed and seasonality.

We have implemented a new parameterization in the manuscript, which builds on a spatially varying transfer coefficient and turning angle (but fixed in time). The results are described in Figures 3, 5 and 6. A parameterization based on the climatology of ice thickness is analogous to that spatially-varying-but-time-constant parameterization. However, we note that the gradients in locally fitted transfer coefficients are sharper than gradients in the climatological ice thickness, therefore, the reduction in the spatial error is higher for the parameterization based on local fits of alpha rather than the thickness climatology. The following supporting figure shows maps of the magnitude of the transfer coefficient for local fits (left) vs a parameterization based on the thickness climatology (right).

In short, I invite the authors to reflect on the trade-off between the improved performance of their free-drift model vs the added complexity it brings, noting that PIOMAS' velocities (or from alternative ocean/ice models) are as available as their sea-ice thicknesses.

**Other comments:**

It seems the importance of sea-ice motion tracking is described in two blocks, first L39-44, then L116-123. Maybe combine these two blocks (or make it clearer why they stand apart). In these justifications: the Lagrangian tracking of sea-ice age is a key use for year-round sea-ice drift data. *Thank you for the suggestion, we have combined the two paragraphs describing the relevance of sea ice tracking (including a reference to sea ice age calculations) and the need for reliable free drift estimates that cover the summer shoulder season. See L40-54 in the revised manuscript.*

L130-135: What is the justification for not using EASE2? If the justification is to stay close to the NSIDC PathFinder grid, please mention it.

Exactly. We are using the original EASE-grid to stay close to the current distribution of the NSIDC Pathfinder data; in addition, our different Lagrangian sea ice tracking utilities are built with the original EASE-grid definition, so we chose continuity. The following sentence has been added to the text.

L138-140: 'We work with the original EASE-grid -- as opposed to the more recent EASE-grid 2.0 - to maintain continuity with current products distributed by the National Snow and Ice Data Centre (NSIDC) and other sea ice tracking utilities that were built on the original definition of the EASE-grid.'

L165: Andersen et al. 2007 did not look at the NSIDC CDR (it had not been released). Kern et al. 2019 and Kern et al. 2020 do evaluate the NSIDC CDR V3 (among 9 other CDRs).

Kern, S., Lavergne, T., Notz, D., Pedersen, L. T., Tonboe, R. T., Saldo, R., and Sørensen, A. M.: Satellite passive microwave sea-ice concentration data set intercomparison: closed ice and ship-based observations, The Cryosphere, 13, 3261–3307, https://doi.org/10.5194/tc-13- 3261-2019, 2019.

Kern, S., Lavergne, T., Notz, D., Pedersen, L. T., and Tonboe, R.: Satellite passive microwave sea-ice concentration data set inter-comparison for Arctic summer conditions, The Cryosphere, 14, 2469–2493, https://doi.org/10.5194/tc-14-2469-2020, 2020.

Thank you for bringing this to our attention. We modified the text to reflect the recent findings of Kern et al.

L169-173: 'Earlier reports by Anderson et al. (2007) on different types of passive-microwave concentration retrieval indicate errors of ~5% in the winter. More recently, NSIDC-CDR specific investigations by Kern et al. (2019,2020) find a slight bias-high of +1% to +3.5% with a standard deviation of ~5% for winter-type (~100% ice concentration) ice conditions ; while in the summer, the error is found to be much larger, with a standard deviation of 35%, and a mean overestimation on the order of 5 to 10%, when compared to MODIS-derived concentration.'

Section 3.2: The methodology section was not very clear and did not contain all the methodologies involved. I suggest to:

• Move what currently is at the start of section 4 (L246-256) into section 3; *Edited as suggested.*

• Also include the parametrisation on sea-ice thickness in section. The parameterization for a thickness-dependent \alpha is included in Table 1. The detailed description of this parameterization appears in the Results section 4.2, because it builds up on the maps of the spatial distribution of the transfer coefficient which are also described in section 4.2.

• Move L224: "Note that ..." to when the 1st velocity statistics are commented in section 4. *Edited as suggested.*

• Clarify what you mean with L226-230. Is this really needed for the iterations to converge (in my experience with the free-drift model, it converges fast without those steps).

Yes. We need to perform a small number of 'outer loop' iterations (i.e.: alternating between solving for the coefficient alpha and the currents Uw), since updated values of alpha will slightly modify the value of the Uw currents in the next iteration, and vice-versa. It does converge fast (usually within 3, max 4-5 of these outer loops). One could also set up the problem differently and solve for everything in a single least-squares operation, but the way by which the MATLAB lsqcurvefit function takes its input, it is easier to run with an outer loop (given that we search for a global alpha and local Uw's). When the currents are ignored, as for the parameterization alpha\_0, this iterative process is not required and the problem reduces to a single call to the least-squares solver.

Section 4: Methodology of the validation: did you use a circular-statistics version of the RMSE for the angles? If yes, specify it. If not justify why not.

We thank the reviewer for bringing this item to our attention. We now use the circular-statistics proposed by the reviewer. The new results are included in Table 1 of the revised version of the manuscript. The numbers do not vary very much, which we attribute to the concentration of the directional error distribution around zero.

 $\begin{aligned} \alpha_p : \Delta \theta &= -9.4 \pm 49.2^{\circ} (before: -6.9 \pm 54^{\circ}) \\ \alpha_o : \Delta \theta &= 2.4 \pm 49.4^{\circ} (before: 2.3 \pm 53^{\circ}) \\ \alpha_w : \Delta \theta &= 2.5 \pm 45.7^{\circ} (before: 2.6 \pm 51^{\circ}) \\ \alpha_h : \Delta \theta &= 2.5 \pm 45.6^{\circ} (before: 2.6 \pm 51^{\circ}) \end{aligned}$

L266: "are further reduced"? Here we are in a chain of reductions expressed as %age and it should be clear if the reference is the initial state, or the previous step.

*We have edited the results section (specifically sections 4.1, 4.2 and 4.3) in order to clarify the chain of error reduction.*

L271-273: this sentence about ERA-Interim results was not very clear, and interrupts the flow of ERA5-based results. Maybe re-formulate and move elsewhere? *The sentence was moved after the paragraph, as a short note, which now reads:*

L288-290: 'We note that the use of 10 m winds from the 1/4° ERA5 reanalysis reduces the root-mean-square error on the drift speed from 5.7 cm/s to 5.4 cm/s when compared with the 1° ERA-Interim reanalysis [results not shown].'

L312: "and the net wind-ice ..." would it be easier to state the value of Beta? The reader [expects] Beta but it is not directly given.

This was an omission. The description of the coefficients now reads as:

L362-364: 'The error minimization routine yields an alpha\_h coefficient of 2.0% and a beta coefficient of 0.17 m^-1, which translates into a net wind-ice transfer term that decreases linearly from 2% to 0% when sea ice reaches a thickness of 6.1 m.'

L322-326: what seasonal cycle did Thomas (1999) achieved with his monthly tuning? If this is not known, can we hypothesize?

We can hypothesize that Thomas (1999) obtains a seasonal cycle similar to what we show in the present manuscript. Unfortunately, Thomas (1999) does not explicitly show the spatially-averaged drift speed seasonal cycle for observations and for the different models presented in their paper. Nevertheless, we can infer this information from figure 17a of their paper, which displays the seasonal cycle of the magnitude of the wind-ice transfer coefficient (alpha term), which is calculated by breaking down the buoy and wind data in monthly bins. The coefficient is at a minimum in the winter (Feb-Mar-Apr), and increases through the summer until its peak in September, after which it decreases back toward its winter value. This is inversely proportional to the seasonal cycle for ice thickness (maximum thickness in the mid-winter and minimum in September). Since our parameterization for the wind-ice transfer is based on ice thickness, we can reasonably suppose that our seasonal cycle for alpha (averaged over the time series) is similar to that of Thomas (1999). Achieving similar seasonal cycles - using different approaches - strengthens our confidence in the underlying process that we highlight in the manuscript!

L370-371: ice-ice interactions are a possible explanation. But also the functional relationship from dX,dY to direction is highly non-linear for small velocities, hence small errors in dX,dY will bring high errors in the direction. Maybe refer to Stoffelen 1998 (Appendix B) for these non-linear relationships and their impact on error propagation?

Stoffelen, A. (1998), Towards the true near-surface wind speed: Error modeling and calibration using triple collocation, J. Geophys. Res., 103, 7755–7766.

Thank you for this reference. We expanded on the description of directional errors at low sea ice and wind speed. The following sentences were edited/added.

L439-446: 'This result is expected for two reasons. First, instances of low sea ice drift speed, when not driven by low winds, can be the result of important internal ice stresses. Since internal ice stresses are not considered in the free drift momentum balance, the direction of free drift estimates over regions

subject to coastal effects or populated with thick, compact ice will be different from buoy observations. Second, nonlinear behavior becomes more important at low wind and ice velocity. The model error for wind itself depends on the speed, and important errors can result from small directional errors at low wind speed (Stoffelen, 1998). In addition, the momentum balance for sea ice exhibits a stronger nonlinearity at very low ice drift speed, and the assumptions that were made initially when posing a linear free drift model do not hold either (Thorndike & Colony, 1982).'

L401: Please add a reference for the pre/post 2000 breakpoint in the ice trend. This break is not obvious from the sea-ice extent data.

We present results for pre/post 2000 to split the time series in two equal parts. Coincidentally, it is also supported by the breakpoint analysis of Goldstein et al. (2018), who find significant breakpoints around 1998/2000/2001/2002 depending on the region investigated. The opening of the paragraph now reads:

L463-465: 'We produce estimates of the surface currents before and after 2000 (Figure 9a and 9b). We choose to split the data (spanning 1979-2019) around the year 2000 in order to obtain two periods of roughly equal lengths; this choice is also consistent with breakpoint analyses that indicate structural shifts in sea ice area near the turn of the century (Goldstein et al. 2018).'

Goldstein, M. A., Lynch, A. H., Zsom, A., Arbetter, T., Chang, A., & Fetterer, F. (2018). The step-like evolution of Arctic open water. Scientific reports, 8(1), 1-9.

L445-458: Here you discuss the differences between your currents and those from the other sources. You bring twice that the sea-ice rheology in the modelling framework might be too approximated, leading to errors in their currents. However, your own estimates neglect ice-ice interactions altogether. How can you be sure that the issue is on the ocean/ice modelling side and not on your side? Maybe revise to bring more balance in this discussion.

Thank you for this suggestion. We edited the text in order to provide a more balanced discussion about the discrepancies between the different reanalyses and our retrieval of the ocean currents, and acknowledge limitations of our methodology. This section now reads as:

L514-520: 'The low speed bias north of the Archipelago can be due to a misrepresentation of the internal ice stress, both in the models and in free drift estimates. We note that our method tends to overestimate fast ocean currents, and underestimate slow currents. The atmospheric-driven ice motion is generally overestimated in regions with little-to-no sea ice motion (Figure 6), which consequently drives the ocean current estimate down in an attempt to balance the observations. On the other hand, our parameterization imposes a cap on the transfer coefficient alpha\_h, therefore, in regions where ice drifts the fastest, an underestimation of the ice drift speed can drive the associated current estimate upward.'

L487: This is maybe where one could balance the improvements brought by the new methods with its added complexity.

We have added a sentence acknowledging the reliance on ice thickness from a renalysis. L555-556: 'The proposed parameterization relies on sea ice thickness from the PIOMAS reanalysis that is produced in near-real time.'

We do not consider the added cost to be a fundamental drawback, since the objective here was to recalculate free drift estimates over the observational period, for which this data is readily available.

Appendix A: it is very commendable that you took the efforts to document what did not work very well.

Thank you!

Acknowledgments: Maybe add acknowledgments for the data providers (C3S, NSIDC, IABP, etc...) *Edited as suggested.*

**Editorials:**

Title: you have "sea-ice" twice. Maybe drop the 1st of them (A new state dependent...)? *Edited as suggested.*

L3: "Building on the fact"  $\rightarrow$  "Observing" *Edited as suggested.*

L4: "a structure as the"  $\rightarrow$  "a structure similar to that of" *Edited as suggested.*

L13 : "mean and root-mean square error": it took me a while to decipher. Maybe "mean error and root-mean square error" is more readable? *Edited as suggested.*

L20: what do you mean by "observations acceleration"? The observed acceleration? *Observed acceleration. This has been edited in the text.*

L53: Lavergne et al. 2010. Check the publication year. *Typo in the manuscript; the publication year has been corrected.*

L154: Consider cite https://rmets.onlinelibrary.wiley.com/doi/full/10.1002/qj.3803 for C3S/ECMWF ERA5 data. *Edited as suggested.*

L161: Given the recent release of their V4, you could specify that you used V3 of the NOAA/NSIDC SIC CDR. *Edited as suggested.*

L384: "use" is maybe not the right word since your ocean currents are a result?

Yes, the ocean currents are a result, but they do feed back into the estimation of the free drift. We have reworded this as 'integration of a fixed ocean climatology'.

L384-391: this discusses the limitations that your ocean currents are fixed over time. But we haven't seen the ocean currents yet. Consider moving this text towards the end of the ocean-current section.

Edited as suggested.

L425: missing space between "the" and "gyre" *Edited.*

L427: "this stabilization" (was described in the previous sentence). *Edited as suggested.*

L437-438: analogous to the seasonal cycle ("to" missing). Similar to... ("to" missing). *Edited.*

L467-469: "This established ... coefficient": this sentence seems broken. *We have combined it to the previous sentence, which now reads as:*

L542-545: 'Free drift is defined from the balance of the surface wind stress and ocean drag; i.e. excluding internal ice stresses, coriolis and sea surface tilt. This results in the standard linear relationship:  $U_i$ =alpha\* $U_a$ + $U_w$  (Thorndike1982, Lepparanta2011), where  $U_i$  is sea ice velocity,  $U_a$  is wind velocity,  $U_w$  is the surface ocean current, and alpha is the wind-ice transfer coefficient."

L489: missing opening parentheses for the citation to Thorndike and Colony *Edited.*

Fig 10, panel g): the transparency of red/blue did not allow to fully observe the two distributions. *Edited.*

**RC2: 'Comment on tc-2021-249', Harry Heorton, 18 Oct 2021**

This paper documents the matching of wind driven free drifting sea ice to observations of ice tethered buoy drift. A number of methodological scenarios are considered. First a constant parameter set for the equations of freed drift are sought. Secondly the ocean surface currents are considered with climatological maps of ocean currents retrieved along with the free drift parameters. Finally the free drift parameters are replaced with sea ice thickness dependent parameters. I suggest this paper for publication. The writing is technically accurate with few typos and grammatical errors. The results shown are compelling and make a useful contribution to sea ice science. In particular it is very interesting to see the wind driven and ice thickness dependent components of sea ice drift quantified. Also the retrieval of ice surface currents and their comparison to other ocean surface data and models is very interesting.

However, I found reviewing the paper quite frustrating and challenging. A few kept parts of the method are lacking, which makes the interpretation of the complex results rather difficult. In particular the description of the Pathfinder data used, and the description of the 'Polar Pathfinder' parameterisation makes many parts of the paper difficult to interpret. The Pathfinder data is described in the data section as pathfinder free drift data, that from the description seems to be a custom made free drift data set using a NCEP/NCAR reanalysis, whilst descriptions of the data in the paper appears to be the version 4 pathfinder dataset I am familiar with, that includes many floe tracking and ice buoy additions to the free drift equations. What data is used here? I suggest finding an alternate description of the base 'control' free drift parameterisation (using the same parameters used to create the Pathfinder dataset) within the paper in order to make it much easier to read.

We have clarified the manuscript as per the reviewer's comments below. With respect to the Polar Pathfinder data, the main Polar Pathfinder data field consists of the merged ice motion vectors that combine buoy+satellite+free drift. In addition to the merged ice motion vectors, each source of sea ice motion is provided individually, referred to as raw motion vectors. Therefore, when we set a baseline based on the 'Polar Pathfinder free drift', we refer only to the free drift estimates calculated by the NSIDC from the NCEP/NCAR atmospheric reanalysis. We reworded the references to the 'Polar Pathfinder free drift input', and have included the following clarification sentences in the data section and in the results section:

L151-154 (Data section 2.3): The Polar Pathfinder dataset consists of merged sea ice motion vectors derived from buoy, satellite, and free drift estimates. The NSIDC also provides the raw motion vectors from each data product. In the following, we use the free drift raw vectors processed by the NSIDC -- equal to 1% of the geostrophic wind speed from the NCEP/NCAR atmospheric reanalysis (2.5° resolution), with a 20° wind turning angle.'

*L271-272: (Results section 4.1): 'We use the NSIDC Polar Pathfinder free drift input data (Tschudi et al. 2019) as a benchmark to evaluate the improvement attributable to other parameterizations.'*

The description of the methods used in section 3.2 requires more information. The paper appears to have three key scenarios where various parameters are retrieved and additional data is also retrieved (ocean currents, wind-ice transfer coefficient in figure 4(a)?). It will be much easier to understand these methods if each can have an equation (7) describing exactly how the values are retrieved. Also in this section more information is required on which data are collected for each case. It is confusing at times how many buoy observations, how many reanalysis based free drift equations and how many parameter values are considered in each scenario. For example are you seeking maps of \alpha and \theta or single values? The case of allowing for variable ocean surface currents is very difficult to understand. Do you actively search of ocean currents with each pixel a free parameter? Or is it simply the residual in the equation as mentioned later in the text.

Number of buoy observations:

• The total number of buoy observations (and co-located winds, and thickness when applicable) used to retrieve the different free parameters is the same for each scenario. There are ~450,000 buoy drift vectors available over the 1979-2019 period (see data section of original manuscript). When optimizing the transfer coefficient, all of the buoy observations are used; however when retrieving the ocean current within each pixel, only buoy observations within a certain radius of that grid cell are used: the spatial density of buoy data can be seen in Figure 1.

**Number of free parameters:**

• There are 2\*n+2 or 2\*n+3 free parameters in each of the scenario; where 'n' corresponds to the number of grid cells where we estimate the ocean current (2\*n because we retrieve the u-and v- components of the ocean current), in addition to 2 or 3 free parameters for the transfer coefficient (alpha, theta, beta), depending on the scenario. The 2-3 free parameters associated to the transfer coefficient are global, i.e. common to the entire Arctic, whereas each of the '2\*n' ocean current free parameter is local.

**Alpha and Theta:**

• We seek single values for alpha and theta (and beta). While it does translate into a single value of the transfer coefficient for the earlier scenarios, the inclusion of sea ice thickness as an input parameter allows for a spatially and temporally varying transfer coefficient. In figure 4a, we present a map of the magnitude of the transfer coefficient in a climatological sense. This map did not serve as an input in the parameterizations of the original manuscript, but upon suggestions from the reviewers we have introduced a new scenario that uses maps of alpha and theta - i.e. spatially varying transfer coefficient and turning angles, but fixed in time (see section 4.2)

**Variable ocean current:**

• We indeed search for an ocean current within each grid cell; it is treated as a free parameter in the least squares fitting routine. The important thing to understand here is that the calculation for each of the ocean current is only performed on a subset of the buoy data, data points that fall in a small search window around the grid cell where we want to produce the ocean current estimate. The ocean current is the value at the origin that minimizes the least squares error in a 2D plot of Ui vs alpha\*Ua.

The points above have been clarified in the revised manuscript (see data and method sections for details).

I suggest the paper for publication after minor revisions. While the paper was difficult to interpret at times, the quality of the figures and results suggest that it will all work out if the points above are addressed.

H. Heorton.

Thank you for the constructive comments.

**Specific points.**

L4-5 'has a structure as the spatial distribution of sea ice thickness' Please reword this sentence as the above is very difficult to interpret *Rephrased to:*

*L3-5: 'Given the fact that the spatial distribution of wind-ice-ocean transfer coefficient has a similar structure to that of the spatial distribution of sea ice thickness, ....'*

L 5 please include to what you have introduced this parameterisation. Which model are you using?

Following up on the sentence just above::

L5-6 '... we take the standard free drift equation and introduce a wind-ice-ocean transfer coefficient that scales linearly with thickness.'

L 13 'minimize' - minimizes *Edited*

L13 what cost function? *The cost function is described in the methodology. We have edited the sentence as follows:*

L14-15: 'The optimal wind turning angle obtained from a least-squares fitting is 25°, resulting in a mean error and a root-mean-square error of +2.6° and 51° on the direction of the drift, respectively.'

It is unclear from the abstract what exactly you have undertaken in this study.

Free drift estimates are important for complementing other sources of ice motion when producing a merged sea ice motion dataset. In this paper, we propose a parameterization for free drift that reduces the error with respect to buoy drift data; and henceforth should also contribute to reducing the error on sea ice velocity in merged ice motion datasets. This information is communicated in the revised abstract as:

'Free drift estimates of sea ice motion are necessary to produce a seamless observational record combining buoy and satellite-derived sea ice motion vectors. We develop a new parameterization for the free drift of sea ice based on wind forcing, wind turning angle, sea ice state variables (concentration and thickness) and ocean current (as a residual). [...] This represents a 35% reduction of the error on drift speed compared to the free drift estimates used in the Polar Pathfinder dataset.

[...] This new dataset is publicly available for complementing merged observations-based sea ice drift datasets that includes satellite and buoy drift records .'

L 25 I enjoyed this quotation. Thank you *Thanks!*

**Section 2.3**

Here you describe a Pathfinder Data set I am unfamiliar with. The citation and DOI given point me towards the Polar Pathfinder version 4 data set. This data is the ice motion data set derived from Floe tracking algorithms and the IABP buoy data set you mention above along with NCEP/NCAR reanalysis. Is the data you are using? The free drift data you mention in this section appears to be a conversion from NCEP/NCAR surface winds through free drift ice drift equations only (using constant parameters). This data is then interpolated onto the 25km EASE grid. Is this correct? Did you create this data yourself? If so can you title this section appropriately. For instance if this free drift data is NCEP/NCAR derived free drifting sea ice estimates, then please call it this.

We now call it the 'Polar Pathfinder free drift input data' in order to make things clearer. The NSIDC provides the raw individual ice motion estimates from different sources alongside their main product, which is the Polar Pathfinder merged sea ice motion vectors that you are familiar with. In this study we take only the free drift estimates as calculated by the NSIDC, which are taken as 1% of the speed + 20° turning angle from the NCEP/NCAR-derived geostrophic winds. The only transformation to the dataset is an interpolation from the 50 km EASE grid to the 25 km EASE grid to match our common data grid.

**Section 3.2**

Equation (7). If I have interpreted the text from line 225 correctly, then  $U_i^{fd}$  in equation (7) is also a function of  $U_w$ . Is this correct? If so is it possible to show this within equation (7)? As in  $U_i^{fd}( \lambda_u)$  ( $\lambda_w$ )? If you use different equations at different times, then perhaps listing all the equations used will aid the reader.

Indeed, the least-square minimization is broken in two: we solve once for \alpha and \theta, and then once for U\_w, and repeat iteratively until convergence. The equation has the same functional form, but since it optimizes different parameters we agree that it is clearer to show them separately. See L246-248 and Equations 7a and 7b in the revised manuscript.

Whilst this section technically well written I find it hard to follow the exact methodology. Evaluation of equation (7) at a single data point is under constrained, so I assume there is a high number of n data points. You mention that U\_w is only varying spatially. What temporal and spatial resolution do you use for the IABP data, the ERA5 data and for the searched for \alpha and \theta? What is a typical value of n? How do you co-locate the temporarily varying IABP data with the static but spatially varying U\_w, and how does this relate to the extracted parameterisations? The results and conclusions suggest that you create spatially varying maps of \alpha and \theta, and then the results suggest you only extract single values for each case, can

you clarify? For discussion of the role of time averaging of drift and wind data in the free drift calculations see Heorton et al. 2019.

Equation 7, number of data points, typical value of n:

- An additional note was added to the text. Now that we have split equation 7 in two, we specify the number of points that are used for finding the \alpha and \theta values, and the U\_w values. This now reads as (L246-249):
- 'In a first step (Eq. 7a), the U\_w's are considered known, and we find the free parameter(s) \alpha and \theta that minimize the error function over all available data (n=457,915). In a second step (Eq. 7b), the wind transfer coefficient(s) are fixed and we solve for the time-constant but spatially varying ocean currents U\_w(x,y). The number of data points that go into the evaluation of each value of U\_w(x,y) varies spatially (see Fig. 1).'

Temporal and spatial resolution of IABP, ERA5 and other input:

• This is discussed in the Data section. We consider the daily averaged values for each of the variables. The common grid is the 25 km resolution EASE-grid. The buoy ice motion vectors have a sub-grid scale resolution but the positions are expressed with respect to the 25 km EASE grid cartesian coordinates. For co-locating buoy points with values from the other fields, we use a nearest-neighbour approach.

*Co-locating temporally variable IABP with static but spatially varying U\_w:*

• Yes, this is correct, the co-location between a buoy ice motion vector and the ocean current estimate is based on position only; irrespective of time since the U\_w represent a climatology. With respect to the least-squares fitting approach, this just means that we have a mix of local and global free parameters: the global free parameters are the \alpha and \theta that allow for a a flexible transfer coefficient that scales with thickness; the local free parameters are the current estimates \U\_w that act as a first order local correction to the wind-driven component of ice motion.

Spatially varying maps of \alpha and \theta:

• \alpha, \theta (and \beta\_h in the scenario that includes thickness) take single values for the entire Arctic. For the first two (static) scenarios, this does translate into a wind-ice transfer coefficient that is identical in all places and all seasons. However, as soon as we introduce a dependence on thickness, we allow for the wind-ice transfer coefficient to vary spatially and temporally. We only prescribe spatially varying maps \alpha or \theta. in the new \alpha\_ij parameterization that was introduced during the revision (see results section 4.2 for more details).

See Heorton et al. 2019:

• Thanks for this reference.

**Section 4,**

I'm struggling to understand what has been performed during each scenario. What data is used in which part of the equation and what is being solved for in each case?

We have moved the description of different scenarios to the methodology section (3.2), immediately above the description of the minimization procedure, hopefully this makes it clearer and improves the flow. The scenarios/parameterizations are described in greater details in the results section, but here as a summary:

- \alpha\_p, Polar Pathfinder free drift input: nothing is done here, we simply take the free drift estimates as provided by the NSIDC.
- \alpha\_0: this one uses a constant wind-ice transfer coefficient and has no currents. Therefore it is only based on buoy data and ERA5 wind data, and requires a single call to the minimization function (Eq. 7a) for finding global and constant values for \alpha\_0 and \theta\_0
- \alpha\_w: the wind-ice transfer coefficient is constant as for the previous scenario, but we now introduce ocean current estimates. Again this is only based on buoy data and ERA5 winds, but it requires a second error minimization step (Eq. 7a + 7b) that solves for local U\_w's.
- \alpha\_ij (new): spatially varying wind-ice transfer coefficient, the turning angle and the U\_w's are computed (a local fit is performed for each grid cell). This is a new scenario that we have introduced during the revision process.
- \alpha\_h: we introduce a new input variable, sea ice thickness from PIOMAS (so we have buoy data, ERA5 winds and PIOMAS thickness that go into the minimization exercise). We call iteratively Eq. 7a and 7b, and find global values for \alpha\_h, \beta\_h, \theta\_h, and local values for U\_w's.

**L 240 So does fig 1 indicate the density of IABP buoy drift data?**

*Yes, exactly! This was not explicitly mentioned in the caption, which we have edited as follows:*

(Figure 1): 'Based on the density of IABP buoy drift data over the 1979-2019 period: spatial map of the number of data points used for the evaluation of the surface currents U\_w at every grid location.'

Table 1, can you expand on what exactly is meant by the mean bias error metrics. Each case has two numbers, which is which? How is the value in brackets for ui and vi all equal to 0.0 for every case except for \sigma\_p. What is this number? Is it the mean dimensional bias for this value? Does this show that the error for this value is evenly spaced around the mean?

We present two error terms for comparing the velocity at a buoy point  $(x_b)$  to our free drift estimate of ice motion  $(x_fd)$ , the root-mean-square error, and the mean error, defined as:

$$RMSE = \sqrt{\frac{1}{n} \sum (x_b - x_{FD})^2}; MBE = \frac{1}{n} \sum (x_b - x_{FD})$$

The RMSE is a measure of the spread of the error around the truth (i.e. buoy drift), whereas the mean error is more indicative of an underlying bias, which is why we prefer referring to it as the 'mean bias error' in this context. In the case of ui and vi, since we have calibrated the parameterization of the components of motion of buoy data, we find as expected a near zero mean bias error: this is just indicative of an error normally distributed around zero for the components of drift, a feature which can also be seen in Figure 3a,b,c (except for the Polar Pathfinder free drift input, which displays a considerable bias on the v-component). To indicate more clearly the difference between the two numbers presented in the table, we have rephrased the table header to '**RMSE** and (Mean bias error)' and also edited the caption which now reads as:

The five rightmost columns present the root-mean-square error (RMSE, bold font), and mean bias error (in parenthesis) on drift speed, u- and v-components, and drift angle, and the explained variance (R^2), evaluated at the IABP buoy data location.'

L 252, Polar Pathfinder parameterisation. This line is difficult to interpret when the form of this data is unclear. If the data here is free drift ice equations applied to NCAR/NCEP are you extracting \alpha and \theta by comparing winds from ERA5 to free drifting sea ice created using NCAR/NCEP? Do you replace the IABP buoy data in equation (7) with the free drift sea ice data set? It is unclear in section 3 and equation (7) how you evaluate buoy data, free drifting sea ice gridded data, and the minimisation of two free parameters at each point, or for the full data set. Or do these cases just imply that you have applied some parameters to colocated IABP drift and ERA5 winds, and then compared this to the free drifting sea ice calculated from NCAR/NCEP winds (the Polar Pathfinder data)? If this is the case, then this is a strange comparison of two different surface wind reanalysis data sets. If this Polar Pathfinder dataset is the floe tracking derived ice drift data from the citation in section 2.3, and not a 'free drift' data set created for this study, then this comparison makes sense.

Things should be clearer now that we refer to the 'Polar Pathfinder free drift input'. Again, we take exactly the product distributed by the NSIDC, which is calculated as  $U_i = 0.01 \text{ exp}(-i 20^\circ) U_a$ , where  $U_a$  is the geostrophic wind based on the NCEP/NCAR atmospheric reanalysis. Also, referring to the Polar Pathfinder free drift input as 'reference' or 'standard' might have been misleading: what we really mean is 'benchmark' in terms of performance against buoys; this has been edited in the text:

*L271-272: (Results section 4.1): 'We use the NSIDC Polar Pathfinder free drift input data (Tschudi et al. 2019) as a benchmark to evaluate the improvement attributable to other parameterizations.'*

For clarity: our reference, or 'truth', is always the IABP buoy-derived daily averaged ice motion. When we optimize the coefficients in the different scenarios, we always co-locate the ERA5 winds (and any other variables) to the IABP buoy drift positions. When we calculate the error metrics, we co-locate either the Polar Pathfinder free drift input, or our home-brew free drift estimates, to the entire set of IABP buoy drifts.

We agree that there is a jump from free drift estimates based on NCEP/NCAR reanalysis winds (NSIDC/Polar Pathfinder) vs ERA5 10m winds (our free drifts), but this is also part of the objective of this project. We want to obtain better free drift estimates than what is being used in the current distribution of the Polar Pathfinder, and upgrading to a more accurate and higher resolution wind reanalysis is also contributing to this. We quantify the contribution of the higher resolution of the wind fields, by comparing free drift forced with ERA-Interim and those forced with ERA5 (see L288-290).

L 253, does constant wind-ice transfer coefficient mean single values of \alpha and \theta were calculated for all the data? Or do you produce static maps of \alpha, \theta?

*Constant means a single value of \alpha and \theta optimized over the entire dataset (this is the case for the \alpha\_0 and \alpha\_w parameterizations).*

We have introduced a new parameterization (\alpha\_ij) with static maps of \alpha and \theta. It makes another intermediate step between a single coefficient and the thickness-based parameterization which changes in space and time. The new parameterization which builds on static maps of \alpha and \theta is described in section 4.2, and new panels have been added to figures 3,5,6 to show the results obtained with this parameterization for free drift.

L 260 Can you comment more of the differences for the \sigma\_p case? Here you are comparing this difference between 'Polar Pathfinder' parameterisation, a new constant parameterisation and the IABP buoy drift vectors? Is this correct? Is this Pathfinder the v4 data set? If this is the case then the changes discussed in this section come from either the parameters in the free drifting ice used in the Pathfinder dataset, the correction to this from floe tracking and buoy drift performed as part of the creation of the Pathfinder v4 dataset, or the use of NCAR/NCEP compared to the ERA5 dataset used in this paper? If however this 'Pathfinder free drift dataset' is of your own creation, then what comes from using a different reanalysis and what comes from changing the parameters?

The Polar Pathfinder free drift input is compared to the buoys, which sets a benchmark, and then we evaluate our other free drift parameterizations against the buoys and observe the improvement. Indeed, different factors are contributing to the error reduction. If we focus on the first step (from \sigma\_p to \sigma\_0 [single values of \alpha and \theta, no currents]), both the combination of using ERA5 over NCEP/NCAR, and calibrating \alpha and \theta directly to the IABP help reducing the error. We don't have the intermediate scenario with NCEP/NCAR winds + calibration on the IABP data. We can nevertheless make an hypothesis. The rms-error is reduced by 0.3 cm/s when using the 1/4° ERA5 winds over the 1° ERA-Interim winds (see L288-290). The total error reduction from \sigma\_p to \sigma\_0 is 1.2 cm/s, therefore a finer wind reanalysis contributes to ~25% of the error reduction in this case. We can suppose that the contribution of the transition from NCEP/NCAR to ERA5 is even larger (ERA performs marginally better in the Arctic), so a conservative estimate would be that 25 to 50% of the improvement in this first step comes from the different reanalysis and the remainder from fine tuning the free drift parameters by performing a calibration directly to the IABP data.

*This has been clarified in section 4.1 of the revised manuscript.*

L 261 In this case (\sgima\_0) is a single parameter extracted for \alpha \theta for all data? Correct, the rms-error and the mean bias error report in that section of the text represent single error parameters for the whole Arctic and for the entire length of the time series. We break down the spatial and temporal variability of the error in Figures 5 and 6.

L 265 Is this case not \sigma\_w? See also figure 3. *Yes, this is a typo and should be read as \alpha\_w. Thanks for catching that!*  L 280 this peak in zero-velocity ice drift suggests that this is the floe tracking Pathfinder drift data set. This confusion in what Pathfinder data you are comparing to is making this interpretation of your results very difficult.

The peak in zero-velocity arises from the IABP buoy distribution. We have rephrased the sentence to make it more explicit:

L293-295: 'A peak at zero velocity is present in the IABP buoy drift speed distribution, which, aside from low winds, can be attributed to the presence of landfast ice, or thick ice north of the Canadian Arctic Archipelago, or more generally, strong internal ice interactions.'

L299 please describe how the value of wind-ice transfer coefficient is calculated here. Is this a map of extracted values of \alpha? Or the ratio  $|U_i|/|U_w|$ ? Do you compare the directional velocity in this ratio, or simply the drift speed? An equation here may be helpful.

Good point. Essentially, we apply the same least-squares minimization procedure as in Eq. 7a, for retrieving the values of \alpha and \theta, but locally. The following sentence has been added to the text:

L321-323: 'Local values of \alpha are obtained using the same least-squares optimization method (Eq. 7a) on co-located pairs of IABP buoy data and ERA5 wind data constrained geographically within a 75 km x 75 km search window centred on each target grid cell.'

L 302 Also of comparison here is Heorton, H.D.B.S. et al. 2019. Thanks for the reference. We are referencing it on L325 and L331 when discussing the range of values for \alpha and the atmospheric over oceanic drag coefficient ratio.

L 323 bias-corrected parameterisation is \sigma\_0? Please check all text descriptions of \sigma\_w and \sigma\_0. Are you including the consideration of ocean currents in figure 5? *Thanks for catching these inconsistencies. The edited sentence now reads as:*

L380-381: 'The benefits of a thickness dependent parameterization \alpha\_h emerges when we consider the seasonal cycle and interannual variability of sea ice drift speed (Fig. 5). All of the other parameterizations follow the seasonality of the wind speed (Fig. 5a). '

L 326-328 Is this sentence referring to the Pathfinder drift data you are comparing to? If this is the case then this is the v4 data including the floe tracking and buoy data. *This refers to the Polar Pathfinder free drift input data; this has been rephrased in the text.*

L 327 are you using time-dependant variables here, or is this a suggestion for future work? Here we refer to sea ice thickness, a time-dependent variable in the sense that it is not a fixed coefficient but rather a dynamically evolving parameter.

L 425 'thegyre' *Edited*  L 428 Please reword the last part of this sentence, it is difficult to interpret *We have reworded the sentence as follows:*

L491-492: 'In a recent study, Armitage2020 argues that the stabilization of the freshwater content is due to an increase in eddy kinetic energy dissipation, attributed to a thinner, more mobile pack ice.'

L 430 What are the buoy data counts for these two cases as you give on line 402 for pre/post 2000?

The buoy data counts are n=239,515 for the July-to-December high drift season amd n=218,400 for the January-to-June low drift season. The numbers have been included to the text on L466.

L 451 Here you point out the limitations of your model in that it does not resolve sea ice rheology. I think it is also worthwhile to discuss the physical limitations in your representation of the ocean surface currents. You previously cite the Meneghello et al. (2018) paper where the inversion of sea ice to ocean stress is described. Do you think your method of resolving ocean currents using wind forced sea ice free drift may only resolve the wind driven (through the sea ice cover) component of ocean surface currents? Can you account for cases where the ocean currents drive ice drift during low wind speeds? What further information may be presented by including geostrophic currents and resolving Ekman currents?

*Yes, we have edited this section to balance the discussion and acknowledge the physical limitations of our method for retrieving the ocean currents. L514-520 now read:*

The low speed bias north of the Archipelago can be due to a misrepresentation of the internal ice stress, both in the models and in free drift estimates. We note that our method tends to overestimate fast ocean currents, and underestimate slow currents. The atmosphere-driven ice motion is generally overestimated in regions with little-to-no sea ice motion (Fig. 6h), which consequently drives the ocean current estimate down in the fitting algorithm's attempt to match the observations. On the other hand, our parameterization imposes a cap on the transfer coefficient  $\alpha_n$ ; therefore, in regions where ice drifts the fastest, an underestimation of the ice drift speed can drive the associated current estimate upward.'

With respect to low wind speed cases, the current is indeed driving sea ice. in the limit of  $U_a=0$ , then  $U_i=U_w$ . So, the climatology of the ocean currents can be understood as an oceanic conveyor belt on which we impose dynamic wind-driven perturbations. Including geostrophic currents or resolving Ekman currents could potentially further improve the estimation of the ocean contribution to ice motion, but this is beyond the scope of this study.

L 459 I think it is important to point out that your ocean surface currents do not resolve the northern, eastward part of the Beaufort Gyre.

We do not understand this comment. Please clarify and we will address it in the next review cycle.

L 470 It is unclear from the methodology where and when and how you created the maps of wind- transfer coefficient that you compare to the sea ice thickness data. *This is now described in more details on L321-323 and L357-358.*

L 488, how do they appear as a residual? Equation 7 and the associated text refer to them as a value to be solved for.

It is true that the ocean currents are not exactly a residual. It is a simplification from our part, for explaining the origin of this value. For a single data point however, the ocean current would actually correspond to the residual (like in Thorndike & Colony 1982). In our experimental setup, the value of the ocean current estimate corresponds very strictly to the 2-dimensional intercept in a plot of \alpha\*U\_a vs U\_i\_buoy. The optimal value is found by minimizing equation 7b on data points geographically located within our area of interest (search window around a grid cell). We have edited the text in order to clarify this methodological nuance.

L 489 please correct the parentheses. *Edited.*

L 493 - 'post-200s' to 'post-2000s' *Edited.*

L 496 It will be more accurate here to describe them as the 'wind driven component of ocean surface currents'

We have rephrased this as 'wind- and ice-driven component of ocean surface currents' on L576.

Figure 3 caption. Here you list the columns as (pathfinder \sigma\_p, free drift \sigma\_0 and thickness dependant \sigma\_h) but the figure labels show the middle column as \sigma\_w for ocean currents. Please re-write accurately.

The middle column in the original manuscript refers to the parameterization with a constant transfer coefficient and ocean currents, \alpha\_w. The arrangement of the figure has been changed in the revised manuscript and the row/column listing has been fixed.

**Figure 4, which case is this map of \alpha calculated for?**

In the original manuscript, the map in figure 4 did nor refer to a specific parameterization, it was only provided to highlight the spatial co-variance with ice thickness. However, during the revision process we introduced a new free drift parameterization, \alpha\_ij, that uses static maps of \alpha and \theta. The map in figure 4 therefore represents the spatially varying \alpha for this new scenario. The \alpha\_ij parameterization is described in section 4.2 and additional plots were included in figures 3,5,6.

Figure 5, It is unclear which data is compared here. Is this the averaged drift only where there is co-located IABP and parameterised ice drift data? If not then this is an unfair comparison. How do you calculated the relative error? Is it part of the use of equation (7)? Also again I'm unsure what the Pathfinder drift data is. There are large differences in these plots between the parameterised ice drifts and the buoy data. Visually your new free drift parameterisations are much closer to the IABP drift data than the 'Pathfinder' parameterisation, whilst the relative error plots show only a slight reduction, can you explain this? What time periods do the data represent? The ERA5 winds used here are daily averages. Are the buoy drifts also daily averages?

**Data comparison:**

• In this figure, we take all IABP drift data, co-locate the different free drift estimates: 1) Polar Pathfinder free drift input (\alpha\_p), 2) free drift estimates with a constant transfer coefficients, and currents (\alpha\_w) and the thickness-dependent parameterization for free drift (\alpha\_h), and then we break down the time series temporally (by month: 5a,c, or by year: 5b,d), and finally we compute standard error metrics (mean drift speed, RMSE between free drift and buoy, etc.).

**Relative error:**

• The relative error is the rms-error divided by the mean buoy drift for that specific month/year (as indicated in the caption). Mathematically, for a buoy drift speed x\_b free drift estimate speed x\_FD:

• Relative error =
$$\frac{\sqrt{\frac{1}{n}\Sigma(x_b - x_{FD})^2}}{\frac{1}{n}\Sigma x_b}$$

• In other words, the relative error shown in Figure 5 is the curve of the rms-error divided by the black curve in 5a,b (mean buoy drift speed).

**Pathfinder data:**

• Same as for the rest of the study, we use the free drift estimates from the NSIDC/Polar Pathfinder package.

*Difference between the curves:*

• We cannot directly take the distance between the curves of \alpha\_p and those of \alpha\_w and \alpha\_h in panel 5a; and associate it to 5c (or from 5b to 5d). The panels at the top only show the pan-Arctic mean for a given temporal bin (it doesn't whos the rms-error); whereas the bottom panels show the rms-error for each of the free drift estimates, divided by the mean of the buoy drift speed. If we look at the August values for instance, we can see that the relative error is ~0.75 for \alpha\_p, and 0.5 for \alpha\_w and \alpha\_h. Given that the mean buoy drift speed is around 9.5 cm/s, this means that the rms-error for \alpha\_p is ~7.1 cm/s, whereas that of \alpha\_w and \alpha\_h is down at 4.7 cm/s. We decide to show the relative error simply because it makes it consistent across the x-axis (month or year), and not dependent on the mean buoy drift speed at a given time. In the revised manuscript, Figure 5 has been expanded to also include the rms-error and the mean bias error.

*Time period and temporal averaging:*

• The data itself represents daily averages (IABP buoy drift is a daily average, free drift estimates represent daily averages). In Figure, the values represent the temporal average of daily-averaged speed, respective to the temporal bin: i.e.: January is the average of all January daily values over the entire record (1979-2019); 1979 is the average of all daily values in that year.

*This has been clarified in section 4.1 and 4.2 of the revised manuscript.*

Figure 7, which case does this figure relate to? Please indicate this in the caption.

*This case refers to the thickness-dependent parameterization. This has been added to the caption in Figure 7.*

Figure 9, can you clarify the definition of the difference plots? Is it (post 2000) - (pre 2000)? *Post-2000 minus Pre-2000. And summer minus winter in Figure 10. This has been clarified in both figures.*

References:

Heorton, H.D.B.S. et al. 2019. Retrieving Sea Ice Drag Coefficients and Turning Angles From In Situ and Satellite Observations Using an Inverse Modeling Framework. Journal of Geophysical Research: Oceans. 124, 8 (2019), 6388–6413. DOI:https://doi.org/10.1029/2018JC014881.

**RC3: 'Comment on tc-2021-249', Anonymous Referee, 8 Nov 2021**

Brunette et al. present a parameterization for the (free) drift of sea ice where alongside the near-surface wind also the sea-ice thickness is considered to yield more accurate estimate for sea-ice drift. They tune the model by an iterative error minimization procedure in which a residual drift field is allowed for and interpreted as the average field of ocean surface currents. The ice drift estimates are substantially more accurate than the simpler free-drift model, and more accurate than an intermediate approach where the ocean current related residuals are taken into account but the thickness information is not used. The approach appears to be very well suited to improve the filling of gaps in ice-drift data sets based primarily on buoy and satellite-derived drift.

I find this manuscript extremely interesting, enjoyed the reading, and learned a lot. The topic considered is highly relevant, the science is solid, and the writing and presentation quality is overall very high. I have mainly three (not fully independent) points where I think some clarification would be helpful, plus a few minor specific comments. Lastly, there are quite a number of small typos, inaccurate grammar, etc., for which I provide just a few examples below. I recommend to go through the whole manuscript carefully to correct such things.

In summary, I clearly recommend to publish this manuscript in The Cryosphere after minor revisions.

Thank you for this positive evaluation of our work and for the thoughtful comments!
* * *
Main points:

In principle you describe quite clearly in the introduction and methods parts that the "wind-ice transfer coefficient" is defined as a coefficient that already involves the ocean drag, that is, reflects the balance between wind and ocean stress rather than reflecting just the air-ice drag. However, in the abstract and also elsewhere in the manuscript I found myself being confused about this. More specifically, I was at first (and from time to time) thinking that the "wind-ice transfer coefficient" should increase with increasing ice thickness as thicker (deformed) ice should tend to be associated with stronger air-ice form drag air-ice. But obviously, thicker (deformed) ice also entails a higher ice-ocean form drag, and if the thickness dependence of the latter dominates, then the "wind-ice transfer coefficient", correctly interpreted by considering the balance of the forces, can decrease with increasing ice thickness. In fact, I'm wondering whether it wouldn't be clearer to use a different term for the "wind-ice transfer coefficient" that better reflects that it's not only about the air-ice drag.

We appreciate this comment! It is true that the \alpha coefficient defined in this study is more than just about the atmospheric drag; it is really an integrated coefficient that builds on both the air drag and the water drag coefficients. We borrowed the term 'wind-ice transfer coefficient' from the literature (it's even shortened to 'wind factor' in some texts) - but this is no reason to limit ourselves!

**We have reworded this as 'wind-ice-ocean transfer coefficient' when defining this quantity, and further on in the text, we simply refer to it as 'transfer coefficient' to make it less verbose.**

Related to the previous point, I'm left somewhat confused about what the thickness-dependence of the "wind-ice transfer coefficient" actually reflects physically. How much of the dependence (coefficient decreasing with increasing thickness) stems from a stronger dependence of the ice-ocean (form) drag on thickness compared to the dependence of the air-ice (form) drag on thickness? And how much is actually rather due to internal ice stresses, which obviously also tend to play an increasing role with increasing thickness? Is it maybe even almost exclusively the latter aspect that is responsible for the dependence, and the form drag does not play a big role as it might depend in a similar way on thickness for both the air-ice and the ice-ocean drag? I think these considerations deserve more discussion.

It's really about the internal ice stresses here. We use sea ice thickness as a proxy for internal ice stresses. Although it's not the sole mechanism at play, to first order, regions with more abundant thick ice (i.e.: north of the Canadian Archipelago) will experience more convergence, larger internal ice stresses, and the resulting ice motion will be slower than a pure free drifting scenario - this is where the negative dependence on ice thickness arises. The purpose of Fig. 4 is to show side by side the spatial distribution of the transfer coefficient and ice thickness, and the strong co-variability in space. The following sentences have been added to the text:

L364-369: 'The negative dependence of \alpha on h\_i indicates that ice thickness acts as a proxy for the internal ice stresses that are not represented in the free drift formulation. To first order, regions populated by thick ice are also experiencing more convergence, larger internal ice stresses, and the resulting ice motion is slower than a pure free drift solution. The inclusion of a dependence on ice thickness therefore allows for departures from the true free drift, via a scalable transfer coefficient. Note that this is different from parameterizations for the air-ice or ice-ocean form drag, which is a function of ice thickness due to higher surface or bottom roughness.'

Lastly, if the thickness dependence of the "wind-ice transfer coefficient" is largely due to the thickness dependence of internal ice stresses, wouldn't that imply that it is misleading to term your approach a parameterization for the "free drift"? Isn't it rather a parameterization for the "drift", without "free"?

Our approach is based on a solution of the sea ice momentum equation under free drift conditions: this is where the linear relationship between sea ice drift, wind velocity and the ocean current comes from. We therefore find it legitimate to present our approach as a free drift solution for ice motion. We nevertheless acknowledge that the introduction of a dependence on sea ice state variables (thickness) is a way to take into account departures from the 'true' free drift state, in order to make an ice motion estimate that is closer to reality. We believe that the text and the addition of the sentences on L364-369 (discussed above) is enough for the reader to appreciate this nuance.
* * *
Specific comments:

Abstract, and throughout the manuscript: Please make even clearer that the "wind-ice transfer coefficient" also includes the ocean drag! (see general comment above) *Edited. It now reads 'wind-ice-ocean' transfer coefficient in the abstract.*

L10-17: Please state the total time period (1979-2019, right?) for which your results have been obtained!

*Edited.* Yes the trends are calculated over the 1979-2019 period.

L14-15: "The residual from the minimization procedure (i.e. the ocean currents)" -> To be precise, shouldn't the "i.e." be replaced by something like "interpreted as"? *This sentence has been rephrased. L15-16: The ocean current estimates obtained from the minimization procedure resolve key large scale features such as the Beaufort Gyre ...'*

L38: "such as" -> which, apart from SIC and SIT? We only use SIC and SIT and state variables. Edited.

L66: "geostrophic wind" -> I suggest to write "surface geostrophic wind" (to make clear it's the SLP-based geostrophic wind; after all, also the geostrophic wind varies with height). *Edited. This is correct, Thorndike and Colony (1982) refer to the surface geostrophic wind.*

L73,74: Do you know why the small value of 1%, which relates rather to the stronger geostrophic winds (as you describe above), is used in combination with the weaker near-surface winds? Was that simply a mistake (you also show that the PP estimates are low-speed-biased)? However, see next point...

L148-149: Here you state (seemingly contradicting the former?) that PP uses geostrophic winds, and a 20° turning angle (although the latter should be more applicable to actual winds?).

The Polar Pathfinder free drift input is based on the surface geostrophic wind derived from the NCEP/NCAR pressure reanalysis. Therefore, the statement in the data section is correct, and in the previous section we clarified that the winds are the surface geostrophic winds (see L153-154).

L201: Why is U\_w the geostrophic and not the actual surface current?

*True, this should be the actual surface current. Equation 4 is the total motion balance and not just the geostrophic equilibrium. This has been edited.*

L228-229: "Assuming that the component of ice motion that is not explained by the winds can be attributed to ice-ocean drag..." - Wouldn't it be more precise to say: "Assuming that the component of ice motion that is not explained by the winds (AND ICE THICKNESS) can be attributed to THE MODIFICATION OF THE ice-ocean drag BY OCEAN SURFACE CURRENTS"...? Regarding the latter, I mean that the ice-ocean drag is in general already contained in the "wind-ice transfer coefficient", but in the baseline only for an ocean at rest.

I'm not sure to follow the reasoning by which the ocean surface currents would be modifying the ice-ocean drags, and consequently the wind-ice-ocean transfer coefficient. The modification of the integrated wind-ice-ocean transfer coefficient is really mediated by ice thickness in our parameterization. We added a note as per your suggestion: '... not explained by the wind (and the influence of ice thickness)...'

Table 1: I would recommend to provide one more digit for the alpha values and angles. *The digits have been harmonized in Table 1.*

Table 1: The mean angle error difference between alpha\_0 and alpha\_p is about 9, although the angle parameter difference is only about 3. Can you explain this discrepancy?

The Polar Pathfinder free drift input is based on geostrophic winds derived from NCAP/NCAR, whereas our parameterization builds on the ERA5 10m winds. Therefore, even if the optimal wind-ice turning angles are relatively close, there might be an underlying difference in the orientation of the source winds...

L284-287: It looks like the differences in R-squared are to a large degree due to the long-term trend rather than the "interannual variability". The letter could be isolated by considering detrended anomalies. In any case, I recommend to clarify that this is mostly about the long-term trend rather than "interannual variations", if I'm not mistaken.

No, when we look at the interannual variability time series (Figure 5b), neither the Polar Pathfinder free drift input data (\alpha\_p) nor the parameterization with a constant transfer coefficient and non-zero currents (\alpha\_w) capture a long term trend, so in this case the  $R^2$  has to be associated with the explained fraction of the interannual variability. When we introduce the dependence on ice thickness, the parameterization captures much better the long-term trend, and this is reflected in the explained variance:  $R^2=0.76$  for the thickness-based parameterization, vs  $R^2=0.28$  for the Polar Pathfinder free drift input. This is discussed in the result sections 4.1,4.2 and 4.3.

L330-332: "Highest relative errors are expected in the winter, since the ice speed is at a seasonal minimum, and the root-mean-square error is maximal due to wintertime ice interactions not being represented by a free drift model." - I fully agree, but shouldn't it be mentioned that your approach - by including sea-ice thickness - accounts for some of the rheological affects in a very simple way (and thus is not really a "free-drift" approach, see one of the main points above)? *Yes, we have edited this sentence for it to reflect more accurately the thickness-dependent parameterization. It now reads as follows:*

L389-391: 'Highest relative errors are expected in the winter, since the ice speed is at a minimum, and the root-mean-square error is largest due to wintertime ice interactions not being explicitly represented by a free drift model, but only partially taken into account via the dependence on ice thickness.' L384-391: Here I'm just a bit irritated that you start the section with this discussion-type paragraph rather than putting it after the results in the first place.

Totally agree. This paragraph has been moved at the end of the section about the ocean currents and the results and discussion flows more naturally (L529-537).

Fig. 1: Please mention in the caption for which time period this is. Is it only the common period until 2014? I'm asking because the yellow blob north of Sewernaja Semlja looks a bit like this is due to a single expedition where many buoys were deployed, such as the first months of the MOSAiC expedition, although that would be only 2019? This also made me wonder whether some data thinning at very data-rich places and times would be helpful in order not to overfit to such strongly non-independent data?

Figure 1 presents the density of IABP data over the full 1979-2019 period, this has been edited in the caption. Interesting question regarding the data-rich places. We are not worried about overfitting for the transfer coefficient free parameters since the parameters (alpha, beta, theta) are fitted to pan-Arctic data across the entire time series. However, if the data is very clustered time-wise, it might affect the solution for the ocean current estimate. We can see the same feature (the blob north of Severnaya Zemlya) in the error fields in Figure 6.

Fig. 5 caption: "weighted by the mean buoy drift speed" -> shouldn't it rather be "divided by ..."? *Yes, edited.*

##########

Typos, inaccurate grammar, etc. (incomplete, please check throughout the manuscript):

L13: "The wind turning angle that minimize the cost function is equal of 25°" -> "minimizeS" and "equal to" *Edited*

\_\_\_\_\_

L21: "datasets that includes" -> omit "s" *Edited*

L212-214: Please check grammar.

The sentences have been rephrased as follows on L219-222:

This simple relation explains roughly 70% of the variability in sea ice velocity in the central Arctic (Thorndike & Colony, 1982). Limiting cases include low wind speed, strong Coriolis effect due to thick ice, and non-negligible internal ice stresses (Thorndike & Colony 1982, Bouchat & Tremblay 2014).'

L220: Omit comma. *Edited*

L221: lest-squares -> least-squares *Edited*  L244: contest -> context *Edited*

L252: In the parentheses, are there additional parentheses, a comma, or similar missing? *Edited*

L278: Fig. 3e,f -> should that be Fig. 3d,e? *Yes. Edited.*

L347: "Fig. 5c" -> "Fig. 5d" *Edited*

L359: "fro" -> "for" *Edited*

**Authors: additional edits - November 2021**

**Reference to Toyoda et al. (2021)**

L109-111: 'In their ice-ocean model, Toyoda et al. (2021) obtain a 15-20% error reduction on the sea ice velocity fields by introducing parameterizations for drag coefficients, the ice-ocean turning angle, and the ice pressure parameter.'

**Methodology section:**

L258-259: 'The same procedure is applied to all of the free drift parameterizations, with the exception of the \alpha\_ij parameterization where the minimization problem is solved locally, for all of the \alpha, \theta and U\_w free parameters (see section 4.2).'

Results:

Section 4.1:

L270-290: The first paragraph has been partially rewritten to clarify the chain of error reduction. L304-312: A new paragraph has been added to describe the spatial distribution of the errors, based on additional plots that are included in figure 6.

Section 4.2:

This section has been expanded and broken in two parts.

Section 4.2 in the revised manuscript discusses the new \alpha\_ij parameterization that has been introduced in the study during the revision process.

Section 4.3 in the revised manuscript is now the section that discusses the thickness-dependent free drift parameterization.

**Conclusion:**

An additional sentence acknowledging the new \alpha\_ij parameterization has been added to the conclusion.

L558-562: 'Although a parameterization (aij) that builds on static maps of the transfer coefficient and the turning angle yields the lowest time averaged pan-Arctic root-mean-square error and the lowest mean bias error (spatially), key advantages of the thickness-dependent free drift  $\alpha(h)$  are better representations of the seasonal cycle, th